# Coreness Maximization through Budget-Limited Edge Insertion

## Abstract

The Budget Limited Coreness Maximization (BLCM) problem aims to enhance average user engagement by activating a limited number of connections, i.e., inserting up to $b$ edges to maximize the coreness gain of all vertices in a graph. Due to the cascading feature, we prove the BLCM is NP-hard, APX-hard, and not submodular, meaning greedy sequential edge insertion fails to deliver satisfactory results. As a result, solving BLCM requires combinatorial edge insertion and must face the combinatorial exploration difficulty. This paper proposes the first effective and polynomial-time approach to BLCM. It embeds local combinatorial optimization into global greedy search to boost the benefits of combinatorial optimization while restricting its complexity. Specifically, we propose efficient methods to evaluate the cascaded coreness improvements of two local combinatorial strategies, i.e., when a leader or a group of nodes increase their coreness values via local edge insertion. Note that the key difficulty lies in evaluating the cascading effects. Based on these, we propose three efficient combinatorial edge insertion strategies: (1) Leader-Centric Greedy Insertion (LCGI), (2) Group-Centric Greedy Insertion (GCGI), and (3) a Leader-Group Balance (LGB) insertion. LCGI greedily finds the most influential leader that can produce the highest coreness gain together with its followers. GCGI finds the most influential group that can promote the most coreness gain. LGB combines the two strategies to select edge combinations adaptively. We prove the low complexity of LCGI, GCGI and LGB. Experiments conducted on 13 real-world datasets highlight their practical utility and superiority over existing approaches.

## Keywords

$k$-core, Coreness, Budget Limited Coreness Maximization, User Engagement

## 1 Introduction

User engagement in social networks is a key indicator that reflects user involvement level in the community. How to increase user engagement in a social network using a limited amount of budget is a crucial problem. The $k$-core model extracts subgraphs in which vertices have degrees of at least $k$. It can extract high engagement users[25, 29]. This paper considers the coreness value of a vertex as an indicator of the user's engagement in the social network [21]. Here the coreness of a vertex extends the $k$-core concept, which is the largest $k$ value among the $k$-cores that cover the vertex. A case study on the Gowalla [19] dataset is illustrated in Figure 1, which shows the average user check-ins versus the coreness values of the user (vertex). It is evident that the coreness value exhibits a strong positive correlation with the user engagement.

Activating connections, i.e., inserting edges into the network is the most natural way to improve the vertex coreness. But edge insertion generally requires some costly edge activation operations, meaning it is necessary to consider the cost. Therefore, **Budget Limited Coreness Maximization problem** (BLCM) is important. It aims to insert at most $b$ edges into the network to maximize the sum coreness improvements of all the vertices. Note that this BLCM problem is different from the traditional $k$-core Maximization (KM) problem [8, 30, 36, 37], which only focuses on improving the size of $k$-core for a specific $k$. The BLCM problem is also different from the Anchor Coreness (AC) problem proposed by Linghu et al. [21]. The *"anchor"* concept in AC raises the vertex's degree to positive infinity without actually inserting edges, which is theoretically interesting but is hard to realize in practice.

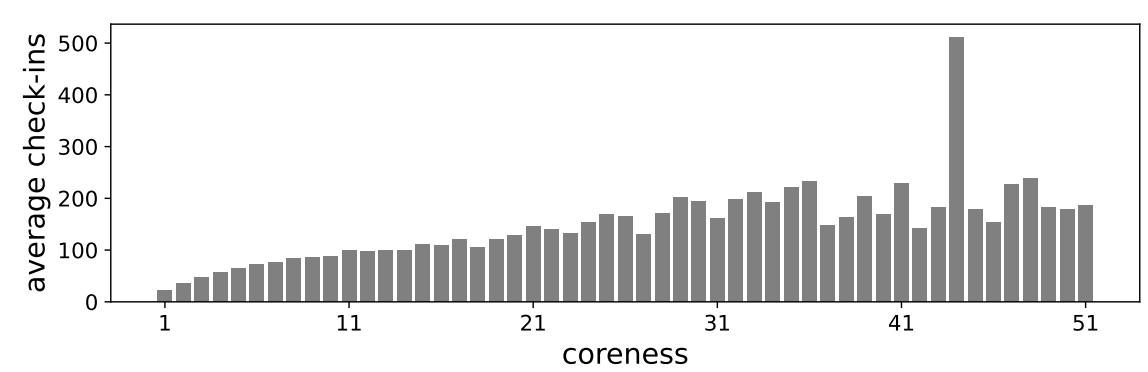

**Figure 1: Vertex coreness vs. User checkins on Gowalla.**

We first prove that the BLCM problem is NP-hard, APX-hard, and that the coreness gain function (the sum of coreness improvements across all vertices) is not submodular. This implies that greedy sequential edge insertion cannot achieve satisfactory results. To address this, we need to explore combinatorial edge insertion, but faces the difficulty of combinatorial explosion. Another difficulty is that a vertex coreness improvement may trigger cascading effect of neighbors' coreness improvements. Therefore, we must on one hand efficiently evaluate each combinatorial insertion, and on the other hand narrow down the search space.

This paper presents the first efficient and effective approach to the BLCM problem. It embeds the local combinatorial optimization into the global greedy search to explore the combinatorial insertion's benefits while restricting its complexity. In particular, we firstly focus on two practical local strategies: (1) increasing a leader's coreness by adding edges, and (2) enhancing a local group's coreness through edge insertion. Both strategies trigger a ripple effect of coreness improvements among the neighboring nodes. We present efficient methods for estimating the potential coreness gain from these cascading effects. For the leader augmenting case, we propose identifying "upstair paths" between the leader and followers to estimate the leader's impact score. For the group augmenting case, we characterize the upstair paths to efficiently estimate the group's impact score.

Based on efficient evaluation of the local combinatorial strategies, the optimized local operations are used as atoms in global greedy search. We firstly propose a Leader-Centric Greedy Insertion (LCGI) strategy, which greedily selects the most influential leader and inserts the corresponding set of edges in each round until the budget is used up. Then, we further propose a Group-Centric Greedy Insertion (GCGI) strategy, which greedily selects the most influential group in each round. We propose group reduction to filter out the poor-performing vertices in the group, followed by group expansion, where the external well-performing vertices are expanded. At last, we present a Leader-Group Balanced Algorithm

(LGB), which chooses LCGI and GCGI adaptively by comparing their scores during greedy selection.

We prove that LCGI, GCGI and LGB have low complexity, i.e., the complexity of LCGI is $O(bmn)$, while the complexity of the latter two is upper bounded by $O(bk_{max}mn)$, where $k_{max}$ is the maximum coreness value of vertices. Since existing methods cannot directly solve the BLCM problem, we compare with other heuristic algorithms and extend the edge $k$-core algorithm (EKC) [36] to solve BLCM. Experiments are conducted on 13 datasets and the results show that: 1) our proposed algorithms are more effective in improving the coreness than other heuristic algorithms; 2) the LGB algorithm has a significant improvement in efficiency while maintaining a good coreness gain compared with the Exact algorithm and the EKC algorithm; 3) the BLCM method has more coreness gain than the vertex-oriented algorithm (VEK) [37] and fast core maximization algorithm (FASTCM+) [30] which are traditional $k$-core maximization algorithms, and the improvement of coreness value is more diverse.

# 2 Problem Definition and Hardness Analysis

## 2.1 Problem Definition

An undirected and unweighted graph is denoted as $G = (V, E)$, where $n = |V|$ represents the number of vertices, and $m = |E|$ represents the number of edges. The neighbors of $v$ in $G$ are defined as $nbr(v, G)$, and the degree of vertex $v$ is denoted as $deg(v, G)$. When the context is clear, we simply use $nbr(v)$ and $deg(v)$ for clarity. Given an integer $k(k \geq 0)$, the $k$-**core** of $G$[26, 28], denoted by $C^k(G)$, is the maximal subgraph of $G$, such that $\forall v \in C^k(G), deg(v, C^k) \geq k$. The **coreness** value of a vertex $v \in V(G)$, denoted by $core(v, G)$, is the largest $k$ such that $v$ is in the $k$-core, i.e., $core(v, G) = max_{v \in C^k(G)}\{k\}$.

When the context is clear, we simply use $core(v)$ as the coreness value of $v$. Through core decomposition [6], the initial coreness of each vertex can be determined, and this process can be accomplished in $O(m)$[3]. Given a graph $G = (V, E)$ and an edge set $D = \{(u, v) : u, v \in V, (u, v) \notin E\}$, the **coreness gain** of $G$ by inserting $D$, denoted by $g(D, G)$, is the total increment of coreness for vertices in $V(G)$, i.e., $g(D, G) = \sum_{u \in V(G)}(core(u, G^*) - core(u, G))$, where $G^*$ is the graph after adding the edge set $D$.

**Problem Statement.** Given a graph $G = (V, E)$ and a budget $b \geq 1$, the budget limited coreness maximization (BLCM) problem aims to find a set $D = \{(u, v) : u, v \in V, (u, v) \notin E\}$ and $|D| \leq b$ such that the coreness gain $g(D, G)$ is maximized, i.e.,

$$D^* = \underset{|D| \leq b}{\arg\max} \sum_{u \in V(G)} (core(u, G^*) - core(u, G)) \tag{1}$$

## 2.2 Hardness Analysis

Theorem 2.1. *The BLCM problem is NP-hard.*

We prove by employing a polynomial reduction from the Maximum Coverage (MC) problem [16] and the rest of the proof can be found in Appendix A.

Theorem 2.2. *The BLCM problem is APX-hard, i.e., for any $\epsilon > 0$, the BLCM problem cannot be approximated in polynomial time within a ratio of $(1 - 1/e + \epsilon)$, unless P=NP.*

We also prove by a polynomial reduction from the MC [16] and the proof is in Appendix A.

Theorem 2.3. *The function $g(., G)$ of coreness gain is not submodular.*

Proof. If $g(., G)$ is submodular, for any graph $G$ and two arbitrary edge sets $A$ and $B$ of $G$, it must satisfy that $g(A, G) + g(B, G) \geq g(A \cup B, G) + g(A \cap B, G)$. But for a graph with $G = (V, E)$, where $V = \{1, 2, 3, 4\}$ and $E = \{(1, 2), (2, 3), (3, 4), (4, 1)\}$, if we set $A = \{(1, 3)\}$, $B = \{(2, 4)\}$, then $g(A, G) + g(B, G) = 0 < g(A \cup B, G) + g(A \cap B, G) = 4$. Therefore $g(., G)$ is not submodular. □

For being NP-hard, APX-hard and not sub-modular, sequential greedy edge insertion cannot provide satisfactory performance to the BLCM problem. It is necessary to explore the combinatorial edge insertion. Combinatorial insertion of $b$ edges into $n^2 - m$ possible positions has obviously $C^b_{n^2-m}$ complexity, which is prohibitive to explore. Another difficulty is that the coreness improvement has a cascading effect, i.e., a vertex's coreness improvement may trigger the coreness improvement of neighbors and further neighbors.

Seeing the difficulties, this paper pursues low complexity polynomial time while effective algorithms for the BLCM problem. The idea is to combine the advantages of local combinatorial search with the global greedy search. To efficiently evaluate the coreness gain of local combinatorial edge insertion operations despite the cascading effect, we focus on two kinds of local combinatorial edge insertion strategies, i.e., adding edges to improve the coreness of a leading vertex, and adding edges to improve the coreness of a group of vertices. We first present methods to efficiently evaluate the coreness gains of these two local combinatorial edge insertion strategies to tailor the cascading effects.

# 3 Gain Evaluation of Local Strategies

We consider two local combinatorial edge insertion strategies. The key is how to efficiently evaluate the coreness gain in each strategy considering the cascading effect.

## 3.1 Leader-Follower Cascading Effect Analysis

We firstly consider the cascading effect that can be triggered by one node, which is called a leader-follower strategy. We denote $core(x) \rightarrow k$ as leading $core(x)$ to $k$, where $k$ maybe $\{core(x) + 1, core(x) + 2, ..., k_{max}\}$ and $k_{max}$ is the maximum coreness value of all vertices in $G$. We call the process of $core(x) \rightarrow k$ as **leading $x$ to $k$-core**, and $x$ is the *leader*. The vertices whose coreness increase accompanying with $x$ are called *followers*, denoted as $F(x, k)$. Then, the leader and followers form a leader-follower structure. **The edge insertion scheme for** $core(x) \rightarrow k$ is to insert edges between $x$ and disconnected vertices in $k$-core until $x$ is promoted to the $k$-core. We evaluate benefit/cost ratio considering all cascadingly triggered followers by inserting edges to **lead $x$ to $k$-core**.

*3.1.1 Leader Impact Score.* Regarding cost, we denote $cost(x, k)$ as the number of inserted edges for $core(x) \rightarrow k$. As for the benefit, it comes from the coreness gain of the leader and the followers. The former is equal to $k - core(x)$, while the latter is equal to the number of elements in $F(x, k)$, i.e., $|F(x, k)|$. By leading $core(x) \rightarrow k$, any vertex $u \in V(G) \setminus x$ can increase its coreness by at most 1, which

is proved in Lemma A.1. So the gain is $|F(x,k)| + k - core(x)$. The $cost(x,k)$ can be calculated by:

$$cost(x,k) = k - |nbr(x,G) \cap C^k| - |nbr(x,G) \cap F_{k-1}(x,k)| \quad (2)$$

where $F_{k-1}(x,k)$ indicates the vertices whose coreness is greater than or equal to $k-1$ among the followers of $core(x) \rightarrow k$. If we want to lead $core(x) \rightarrow k$, at least $k$ edges supported from $k$-core are required. $|nbr(x,G) \cap C^k|$ is the existing support, and $|nbr(x,G) \cap F_{k-1}(x,k)|$ is the followers to be increased to the $k$-core. We select vertices that are not adjacent to $x$ from $k$-core and connect them to $x$. This gets the detailed combinatorial edge insertion scheme to lead $core(x) \rightarrow k$. Then, the impact score of leading $x$ to $k$ can be calculated as:

$$Iscore(x,k) = \frac{|F(x,k)| + k - core(x)}{cost(x,k)} \quad (3)$$

*3.1.2 Leader Followers.* The key to calculating $Iscore(x,k)$ is to calculate $|F(x,k)|$. We present the following theorem which can avoid repeated graph traversal for calculating $F(x,k)$. We denote $I(x,+\infty)$ as the followers after setting $deg(x) = +\infty$ without adding edges, i.e., anchoring $x$, and $I_k(x,+\infty) = \{v | v \in I(x,+\infty) \wedge core(v,G) \geq k\}$, i.e., the followers after anchoring $x$ with original coreness not smaller than $k$. We then prove:

THEOREM 3.1. *For a given graph G, we have* $F(x,k) = I(x,+\infty) \setminus I_k(x,+\infty)$ *for* $\forall x \in V(G)$.

Please see detailed proof in Appendix A. According to Theorem 3.1, our focus narrows to compute $I(x,+\infty)$. But in order to calculate the final $F(x,k)$, we need to calculate the value of $|I(x,+\infty)|$ and the detail elements of $I(x,+\infty)$ to calculate $I_k(x,+\infty)$. To efficiently calculate $I(x,+\infty)$, we find that the vertices belonging to $I(x,+\infty)$ must form a "stair" structure with $x$. Therefore, we define and calculate this "stair" structure by partitioning the graph into shells to calculate the layer structure and the upstair path.

*Definition 3.2.* Given a graph $G$, the $k$-**shell**, denoted by $H_k(G)$, is the set of vertices whose coreness equal to $k$, i.e., $H_k(G) = \{v | core(v,G) = k\}$.

We divide the vertices in $k$-shell into different layers. We set the vertex set of the $j$-th layer in $k$-shell as $H_k^j(G)$, where $H_k^1(G) = \{v | deg(v, C^k(G)) < k+1 \wedge v \in C^k(G)\}$ representing the set of vertices in the $k$-core whose degree is less than $k+1$. The subsequent $j$-th layer is derived from the removal of the preceding $j-1$ layer. Let $G_1 = C^k(G)$, and $G_j$ is the subgrpah induced from $C^k(G) \setminus H_k^1(G) \setminus H_k^2(G) \setminus \ldots \setminus H_k^{j-1}(G)$, then $H_k^j(G) = \{v | deg(v, G_j) < k+1 \wedge v \in G_j\}$, and the layer number $layer(v)$ for $\forall v \in H_k^j(G)$ equals $j$.

Based on the above analysis, we can traverse $k$ from 1 to $k_{max}$ to get the layer number $layer(v)$ for $v \in V(G)$. We call it the **Layer Decomposition** Algorithm. The detailed algorithm pseudocode in the rest of the paper can be found in Appendix B.

*Definition 3.3 (Upstair Path).* A path $x \rightsquigarrow v$ is called an upstair path in $G$ for $v \in V(G)$ if it satisfies the following two conditions: (1) for every vertex $y$ on the path from $x$ to $v$ except $x$, $core(x) \leq core(y) = core(v)$; and (2) for every two consecutive vertices $u_1$ and $u_2$ from $x$ to $v$, it must satisfy $layer(u_1) < layer(u_2) \wedge (core(u_1) = core(u_2))$ or $(core(u_1) < core(u_2))$.

THEOREM 3.4. *If vertex* $v \in I(x,+\infty)$ *, then there must be an upstair path respect to* $x$*, that is* $x \rightsquigarrow v$ *in* $G$.

According to Theorem 3.4, only the vertex that can form an upstair path with $x$ can become an element of $I(x,+\infty)$. We use $CF(x) = \{v | \exists \text{ path } x \rightsquigarrow v\}$ to represent the set of vertices that are candidate followers of $I(x,+\infty)$. Consequently, we only need to traverse the $CF(x)$ vertices to calculate $I(x,+\infty)$. And then we determine the final followers by degree check, which we will describe in Section 4.1.

## 3.2 Group Promotion Cascading Effect Analysis

We further investigate to promote coreness of a local group of vertices and design group score to select the most influential group. We hope to find the groups that can achieve a large increase in sum coreness by inserting a small number of edges in the group.

We find a one-hop structure has above desired property. A vertex may have enough neighbors which can support its coreness improvement, but some of its neighbors lack enough supports to achieve high coreness, so the coreness values of the whole group stay in low values. But only if we add edges among the lack-support neigbhors, the overall group will improve their coreness values. More importantly, such one-hop structure is efficient to identify. We therefore propose to find such one-hop group-centric structure.

*3.2.1 Group Candidates Selection.* To rank the potential groups, we need to identify the group and evaluate the potential coreness gain of the group. The initial step is to identify the central vertex of the group. We firstly define $core^{\geq}(u)$ as the set of $u$'s neighbors whose $core(.) \geq core(u)$. We designate $GC(G)$ as the set of vertices that can become the center of the group, where $GC(G) = \{u | \, |core^{\geq}(u)| \geq core(u) + 1\}$. Given that a vertex $u$ has a higher $|core^{\geq}(u)|$, its failure to achieve $(core(u)+1)$-core is likely due to insufficient degree support among $u$'s neighbors. So, inserting edges among its one-hop neighbors may yield substantial follower returns with relatively low costs.

By selecting a center $u$, the initial group $\mathcal{G}(u) = \{u\} \cup \{v | v \in nbr(u) \wedge core(v) = core(u)\}$. We can find a determined scheme of combinatorial edge insertion in $\mathcal{G}$ to promote the coreness of all vertices in $\mathcal{G}(u)$ to $core(u)+1$. For each $v$ in $\mathcal{G}(u)$, we define $r(.)$ to represent the number of edges required for $v$ to upgrade to $(core(u)+1)$-core. **The scheme to promote** $\mathcal{G}(u)$ is as follows: (1) If $\mathcal{G}(u)$ has $v_1, v_2$, $(v_1,v_2) \notin E(\mathcal{G}(u))$ and $r(v_1) > 0, r(v_2) > 0$, then connect $(v_1,v_2)$. (2) If case (1) doesn't appear, but $v \in \mathcal{G}(u)$ has $r(v) > 0$, then connect $v$ to the unconnected vertices in $(core(u)+1)$-core with $r(v)$ edges. It is clear that this process can promote $\mathcal{G}(u)$ and the number of inserted edges is defined as $Gcost(\mathcal{G}(u))$.

We denote $Ggain(\mathcal{G})$ as the cascaded coreness gain by promoting $\mathcal{G}$. The group score $GS(\mathcal{G}) = \frac{Ggain(\mathcal{G})}{Gcost(\mathcal{G})}$. To calculate $Ggain(\mathcal{G}(u))$, we define $FG(\mathcal{G}(u))$ as the set of vertices whose coreness increases after promoting $\mathcal{G}(u)$, i.e., $v \in FG(\mathcal{G}(u))$ satisfies $core(v,G') > core(v,G)$ and $G'$ represents the graph after promoting $\mathcal{G}(u)$. We define the improved coreness of $v$ as $i_v = core(v,G') - core(v,G)$. Note that $FG(\mathcal{G}(u))$ also includes the vertices in $\mathcal{G}(u)$. Then $Ggain(\mathcal{G})$ is obtained by accumulating the coreness increment of each vertex in $FG(\mathcal{G}(u))$, i.e., $Ggain(\mathcal{G}(u)) = \sum_{v \in FG(\mathcal{G}(u))} i_v$. The key is how to determine the followers.

Theorem 3.5. *If a vertex $v \in V(G)$ is a follower after promoting $\mathcal{G}(u)$, then $\exists x \in \mathcal{G}(u)$, such that $x \rightsquigarrow v$.*

Through theorem 3.5, the search range of the group's followers can be narrowed down to the vertices with upstair paths, which simplifies the calculation process of $Ggain(\mathcal{G})$. Then, we use the improved degree check to determine the final group followers, which will be described in detail in the Section 4.2.

# 4 Greedy Selection of Local Stategies

Empowered by the efficient evaluation of the local combinatorial strategies, we embed the local combinatorial edge insertion into global greedy selection of local strategies to address BLCM both effectively and efficiently. We in particular propose LCGI, GCGI, and LGB three greedy algorithms for local strategy selection.

## 4.1 Leader-Centric Greedy Insertion (LCGI)

This subsection introduces the Leader-Centric Greedy Insertion Algorithm (LCGI). We first focus on how to calculate $I(x, +\infty)$ to get $Iscore(x, k)$. Then, based on the $Iscore(x, k)$, we greedily select the most influential leaders until budget is used up.

*4.1.1 Compute $I(x, +\infty)$.* To calculate $I(x, +\infty)$, we first determine the candidate vertex set to reduce the search range. Then, we judge whether each candidate may have a coreness increase. To find candidate followers $CF(x)$, we denote $cel^{>}(u)$ as the set of $u$'s neighbors whose $core(.) = core(u)$ and $layer(.) > layer(u)$. Similarly, we denote $cel^{\leq}(u)$ as the set of $u$'s neighbors whose $core(.) = core(u)$ and $layer(.) \leq layer(u)$. We utilize a minimum heap $H$ to store the vertices in $CF(x)$ that require traversal. The key for vertex $v$ is defined as $(\text{core}(v), \text{layer}(v))$ for sorting, with $core(.)$ as the first keyword and $layer(.)$ as the second keyword. We start from the leader vertex $x$ and continue to expand the vertices that can form an upsair path with $x$ and put them into the queue $H$.

To further determine whether the vertices in $CF(x)$ belong to $I(x, +\infty)$, we define three statuses for the vertices: *survived*, *unexplored*, and *discarded*. The *unexplored* status indicates that the vertex has not experienced the degree check. On the other hand, *survived* signifies that the vertex has passed the degree check. Lastly, *discarded* indicates that the vertex failed the degree check and will not be traversed in subsequent steps. We set $d^{+}(v)$ as the degree bound of a vertex $v \in CF(x)$, which represents its maximum support at $(core(v)+1)$-core. *Survived* vertices may potentially become *discarded* later if their degree bound is insufficient.

Theorem 4.1. *If a vertex $u \in CF(x)$ satisfies $d^{+}(u) < core(u) + 1$, then $u \notin I(x, +\infty)$.*

*Degree Check.* We use the degree check to check whether the vertices in $CF(x)$ will be in the *discarded* state. We calculate

$$d^{+}(u) = d_{s}^{+}(u) + d_{u}^{+}(u) + d_{>}(u) \tag{4}$$

where $d_s^{+}(u)$ represents the number of *survived* neighbors satisfying $core(v) = core(u)$; $d_u^{+}(u)$ represents the number of *unexplored* neighbors belonging to $\{cel^{\leq}(u) \cap H\} \cup cel^{>}(u)$; and $d_{>}(u)$ denotes the number of neighbors satisfying $core(v) > core(u)$ or $v = x$. According to Theorem 4.1, if $d^{+}(u) < core(u, G) + 1$, $u$ cannot become a member of $I(x, +\infty)$ and is marked *discarded*, which leads to a decrease in the degree bound of $nbr(u)$, which may cause them also to be *discarded*. This discarded state may have a cascading effect and propagate to neighbors again. We summarize the above process and propose the **FindFollowers** algorithm to calculate the set of followers when the coreness of leader $x$ is raised to positive infinity.

*4.1.2 Leader-Centric Greedy Insertion Algorithm.* Based on the calculation of $Iscore(x, k)$, the Leader-Centric Greedy Insertion Algorithm (LCGI) is proposed as given in Algorithm 1. To reduce the search range of $k$, we set $Krange(x)$ as the set of coreness of all vertices in $nbr(x)$ plus 1, representing the coreness values to which the $nbr(x)$ can be increased. We first calculate the $Iscore(v, k)$ for each vertex $v \in V(G)$ and the corresponding $k \in Krange(v)$. Based on the score, we select the leader with the best $Iscore(v, k)$ in each round and insert the corresponding edges into $G$ according to the local edge insertion strategy.

**Algorithm 1:** LCGI $(G, b)$

**Input** : A graph $G = (V, E)$, a budget $b$
**Output** : $D$: the set of inserted edges
1 cost = 0, $D \leftarrow \emptyset$;
2 **while** $cost \leq b$ **do**
3     Layer Decomposition($G, k_{max}$);
4     $epoch_cost = 0, epoch_score = 0, epoch_D = \emptyset$;
5     **foreach** $v \in V(G)$ **do**
6         $I(v, +\infty) = FindFollowers(v, G)$;
7         **foreach** $k \in Krange(v)$ **do**
8             $F(v, k) = I(v, +\infty) - I_k(v, +\infty)$;
9             compute $cost(v, k)$ according to Equation 2;
10            $Iscore(v, k) = \frac{F(v,k)+k-core(v)}{cost(v,k)}$;
11            $D(v, k)$ = Randomly select $cost(v, k)$ vertices (not adjacent to $v$) from $k$-core;
12            **if** $Iscore(v, k) > epoch_score$ **then**
13                 **if** $cost + cost(v, k) > b$ **then** continue;
14                 $epoch_score = Iscore(v, k)$;
15                 $epoch_cost = cost(v, k)$, $epoch_D = D(v, k)$;
16     $cost+ = epoch_cost$, $D = D \cup epoch_D$, update $G$;
17 **return** $D$;

In Algorithm 1, in each round, we first calculates the layer decomposition (Line 3). Then, we traverse all vertices to examine all possible $k$ values and calculate the $Iscore(v, k)$ (Lines 5-11). We sets $D(v, k)$ to represent the connecting edges required from leading $core(x) \rightarrow k$. Simultaneously, if $Iscore(v, k)$ is higher than the previous score (Line 12) and the $cost(v, k)$ is no more than the remaining budget (Line 13), we record it as the current optimal solution (Lines 14-15). Finally, we updates the current cost, and inserts the corresponding edges combination (Line 16).

Theorem 4.2. *The time complexity of Algorithm 1 is $O(b \cdot n \cdot m)$.*

Proof. First, the time complexity of $FindFollowers(v)$ is $O(m)$ as edge traversal occurs at most three times, we will explain this in detail in Appendix B.2. So the time complexity of Line 6 is $O(b \cdot n \cdot m)$. Secondly, when calculating the $F(v, k)$, we only need to traverse $I(v, +\infty)$ once to get all $F(v, k)$ for each $k \in Krange(v)$, so the time complexity of Line 8 is $O(b \cdot n \cdot |I(v, +\infty)|) < O(b \cdot n \cdot n)$. Thirdly, the time complexity of Line 9 is $O(b \cdot k_{max} \sum_{v \in V(G)} nbr(v)) = O(b \cdot$

$k_{max} \cdot m$). Combined with the above analysis, the time complexity is $O(b \cdot n \cdot m)$. □

## 4.2 Group-Centric Greedy Insertion (GCGI)

This subsection introduces the greedy algorithm based on group cascading effect. We first perform GroupReduction and GroupExpansion on the original group to obtain a more efficient group. Then we calculate the group score $GS(\mathcal{G})$ corresponding to each promoted group and select the most influential group in each round.

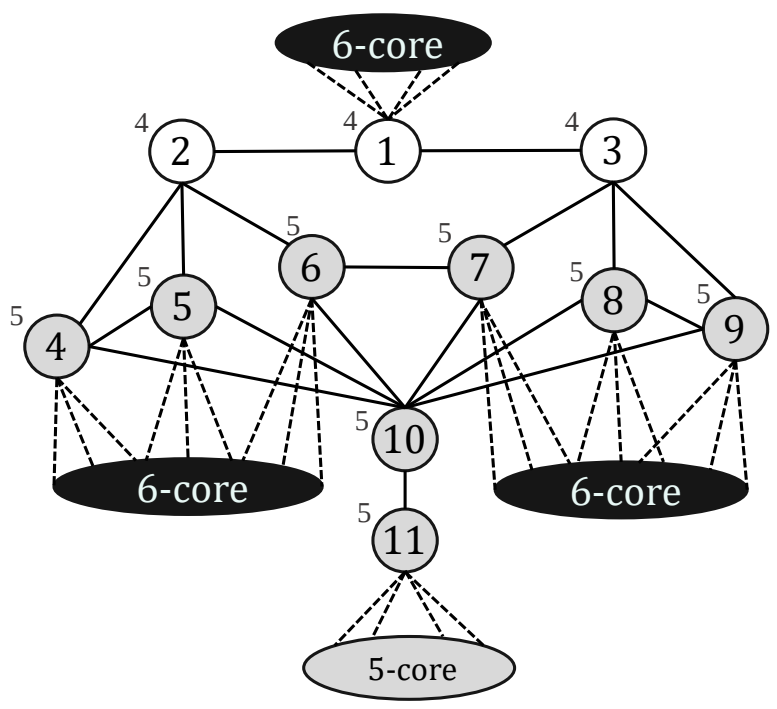

**Figure 2: An Example of core decomposition. The number besides each vertex indicates its coreness value, while the dotted line indicates an edge connected to a vertex in $k$-core.**

*4.2.1 Group Reduction.* The purpose of GroupReduction is to eliminate the vertices with low contributions. For $\mathcal{G}(v)$, we define $K_R = core(v) + 1$. We denote $gc(u) = |\{w | w \in nbr(u) \wedge w \in \mathcal{G}(v)\}|$ as the contribution of vertex $u$ to the $\mathcal{G}(v)$, and $gr(u) = K_R - |N_X(u)|$ as the additional cost brought by vertex $u$, where $X$ denotes the set of $u$'s neighbors and each neighbor $w$ satisfies $w \in \mathcal{G}(v)$ or $core(w) \geq core(u) + 1$. Then, we define the value of vertex $u$ to $\mathcal{G}(v)$ as $gv(u) = gc(u) - gr(u)$.

$gv(u) < 0$ indicates that the contribution of vertex $u$ to $\mathcal{G}(v)$ is less than the cost it brings, so vertex $u$ should be removed from $\mathcal{G}(v)$. The removal of vertex $u$ will alter the $gc(.)$ and $gr(.)$ values of $nbr(u)$ in the group, and we subsequently need to determine whether these neighbors should also be removed from the $\mathcal{G}(v)$. We finally summarize the above process into the **GroupReduction** algorithm, which reduces $\mathcal{G}(v)$ until $\forall u \in \mathcal{G}(v), gv(u) \geq 0$.

*Example 4.3.* In Figure 2, we calculate $gr(11) = 6 - |\{10\}| = 5$, $gc(11) = |\{10\}| = 1$, thus $gv(11) = gc(11) - gr(11) = -4$. Therefore, vertex 11 will be removed from $\mathcal{G}(10)$. And $gr(10)$ and $gc(10)$ are also modified accordingly.

*4.2.2 Group Expansion.* Group Expansion aims to identify vertices outside the $\mathcal{G}(v)$ but making significant contributions to promote $\mathcal{G}(v)$. We define $K_E = core(v) + 1$ and evaluate vertices within one-hop of $\mathcal{G}(v)$. We denote $ec(u) = |\{w | w \in nbr(u) \wedge w \in \mathcal{G}(v) \wedge gr(w) > 0\}|$ as the contribution to the group, and $er(u) = K_E - |N_X(u)|$ (similar to $gr(u)$) as the additional required expenses. Then we define the value of vertex $u$ to the group as $ev(u) = ec(u) - er(u)$. $ev(u) \geq 0$ indicates that adding vertex $u$ can contribute to the $GS(\mathcal{G}(v))$. Therefore, $u$ can be added to $\mathcal{G}(v)$. If vertex $u$ is added to $\mathcal{G}(v)$, then $w \in nbr(u)$ outside the group will change the $ev(w)$ and can be evaluated whether they should be added to $\mathcal{G}(v)$. We summarize the above process into the **GroupExpansion** algorithm, which searches for vertices around $\mathcal{G}(v)$ where $ev(.) > 0$ for expansion.

*Example 4.4.* In Figure 2, we traverse the neighbors within the one-hop range of $\mathcal{G}(10)$ and obtain the set $\{2, 3, 11\}$. Then, we calculate $er(2) = 6 - |\{4, 5, 6\}| = 3$ and $ec(2) = |\{4, 5, 6\}| = 3$, thus $ev(2) = 0$. Similarly, we get $ev(3) = 0, ev(11) = -5$. Therefore, 2 and 3 are added to $\mathcal{G}(10)$ during the GroupExpansion process.

*4.2.3 Group Score.* According to the above two subsections, we have obtained all groups. For each vertex in the group, they must have one of $er(.)$ or $gr(.)$, which we will uniformly record as $r(.)$ same as Section 3.2.1. The $Gcost(\mathcal{G})$ in $GS(\mathcal{G})$ can be easily derived from the $r(.)$ value and the rules of promotion, so the key is in how to calculate $Ggain(\mathcal{G})$. Further, according to the analysis in Section 3.2.1, we need to calculate $FG(\mathcal{G}(u))$.

**Computing** $FG(\mathcal{G}(u))$**.** To calculate $FG(\mathcal{G}(u))$, we define $CF(\mathcal{G}(u))$ as the set of candidate followers. According to Theorem 3.5, $CF(\mathcal{G}(u))$ is the set of vertices that can form an upstair path with $\mathcal{G}(u)$.

Then, we search for the final followers and its coreness increment in $CF(\mathcal{G}(u))$. We set the priority queue $H$ and set the key to $(core(.), layer(.))$. When searching for $CF(\mathcal{G}(u))$, we introduce a hierarchical search, meaning that each hierarchy only explores the vertices with same coreness. The vertices within the initial $\mathcal{G}(u)$ are enqueued into the queue $H$ and then expanded to find $CF(G(u))$. To evaluate whether the vertices in $CF(\mathcal{G}(u))$ will be added to $FG(.)$, we define multiple states for the vertices in $CF(G(u))$. Different from leader's followers, there are cases where the vertex coreness increases to more than 1. Hence, vertices that pass the degree check need to be re-enqueued when the coreness of the hierarchy search increases. We add the *reexplored* status, indicating that it has passed the degree check in the previous hierarchy but has yet to pass it for the current hierarchy. And we save the *survived*, *unexplored*, *discarded* status same as the LCGI. We initialize the array **Survive(.)** to record the current increment in its coreness. Furthermore, note that for the *reexplored* vertices, we default them to the 0-th layer of the high-level $k$-core. This enables us to utilize the concept of the upstair path to continue expanding the vertex in the high-level $k$-core.

*Degree Check.* To determine the vertex state in $CF(\mathcal{G}(u))$, we implement degree check. We denote $core^*(v) = core(v) + Survive(v)$ as the coreness value $v$ may achieve after promoting $\mathcal{G}(u)$. We calculate:

$$d^+(v) = d_s^+(v) + d_u^+(v) + d_r^+(v) + d_>(v) \tag{5}$$

where $d_s^+(v)$ represents the number of *survived* neighbors $w$ satisfying $core^*(w) > core^*(v)$, $d_u^+(v)$ represents the number of *unexplored* neighbors $w$ satisfying $(core(w) = core^*(v) \wedge layer(w) > layer(v))$ or $w \in H$, $d_r^+(v)$ represents the number of *reexplored* neighbors $w$ satisfying $core^*(w) = core^*(v)$ and $d_>(v)$ denotes the number of neighbors $w$ satisfying $core(w) \geq core^*(v) + 1$ or $w \in \mathcal{G}(u)$.

Theorem 4.5. *If a vertex* $v \in CF(\mathcal{G}(u))$ *satisfies* $d^+(v) < core^*(v)$ $+1$*, then* $core(v, G_{\mathcal{G}(u)}) \leq core^*(v)$*, where* $G_{\mathcal{G}(u)}$ *is the graph after promoting* $\mathcal{G}(u)$*.*

According to Theorem 4.5, we traverse the vertices in $CF(\mathcal{G}(u))$ and discard the vertices that can not pass the degree check, and update the $d^+(.)$ of their neighbors, and then judge again whether the updated neighbors will be set as *discarded*. Otherwise, if vertex $u$ passes the degree check, it is defined as the *survived*, and $Survive(u)$ is increased, indicating that $core(u)$ may be increased and $u$ will continue to perform degree check as the hierarchy increases. Based on the above analysis, we propose the **GroupGain** algorithm, which calculates the total coreness gain of promoting $\mathcal{G}(u)$.

*Example 4.6.* In Figure 2, initially, the vertices in $\mathcal{G}(10)$ are added to $H$. During the hierarchical traversal of $H$, the 4-core is first traversed. Vertex 2 is dequeued, and its neighbor vertex 1 (where vertex $1 \in cel^{>}(2)$) is enqueued. We calculate $d^+(1) = d_>(1) = 6 > core(1) + 1 = 5$. Consequently, vertex 1 is marked as *survived*, and $Survive(1) = 1$. Later, during the hierarchical traversal of the 5-core, vertex 1 is enqueued again, and its status is set to *reexplored*. We recalculate $d^+(1) = d_>(1) = 6 > core(1) + Survive(1) + 1 = 6$. Therefore, $Survive(1) = 2$, and it is marked as *survived* once more. Finally, the coreness of vertex 1 is increased by 2.

**Algorithm 2:** GCGI ($G$, $b$)

```
   Input  : A graph G = (V, E), a budget b
   Output : D: the set of inserted edges
1  cost = 0, D ← ∅;
2  while cost ≤ b do
3      epoch_cost = 0, epoch_score = 0, epoch_D = ∅;
4      compute GC(G) ;
5      foreach v ∈ GC(G) do
6          D(𝒢(v)) ← ∅, Gcost = 0;
7          Initial 𝒢(v);
8          GroupReduction(G, 𝒢(v));
9          GroupExpansion(G, 𝒢(v));
10         Ggain=GroupGain(G, 𝒢(v));
11         Q ← {u|u ∈ 𝒢 ∧ r(u) > 0};
12         while ∃x, y ∈ Q ∧ (x, y) ∉ E(G) do
13             Add (x, y) into D(𝒢(v)), Gcost++;
14             r(x) = r(x) − 1,r(y) = r(y) − 1;
15         foreach ∃x ∈ Q do
16             Gcost+ = r(x), update D(𝒢(v));
17         GS(𝒢(v)) = Ggain / Gcost;
18         if GS(𝒢(v)) > epoch_score then
19             if cost + gcost > b then continue;
20             epoch_score = GS(𝒢(v));
21             epoch_D = D(𝒢(v)), epoch_cost = Gcost;
22     cost+ = epoch_cost, D = D ∪ epoch_D, update G;
23 return D;
```

*4.2.4 GCGI Algorithm.* The generation of the group and the calculation of $GS(.)$ have been described above. Based on $GS(\mathcal{G})$, we propose GCGI Algorithm for BLCM problem, as shown in Algorithm 2. It focuses on the computation of $GS$ and the selection of the best group. Each part of the code has been explained earlier. We detail the calculation process of the group cost in Lines 11-17, where $Q$ is derived from the vertex set whose $r(v) > 0$ in $\mathcal{G}(v)$. We select the optimal group by comparing $GS(.)$ and retain the edge combinations that need to be inserted (Lines 18-21).

Theorem 4.7. *The time complexity of GCGI is $O(b \cdot k_{max} \cdot n \cdot m)$.*

Proof. The time complexity of the Algorithm 2 is bounded by GroupReduction, GroupExpansion, and GroupGain. In the worst case, GroupReduction invokes GroupShrink for each vertex, leading to a time complexity is $O(\sum_v deg(v)) = O(m)$. The worst time complexity of GroupExpansion is $O(n \cdot k_{max} + m)$. It is bounded by the maximum number of vertex visits and updates to $er(.)$ and $ec(.)$, as detailed in Appendix B.5. Since the number of times a vertex $u$ is put into $H$ is bounded and does not exceed $core(u)$, the worst time complexity of GroupGain is $O(m \cdot k_{max})$. Since the above three algorithms are called at most $b \cdot n$ times, the time complexity of Algorithm 2 is $O(b \cdot n \cdot m \cdot k_{max})$. □

## 4.3 Leader-Group Balance Algorithm

We combine the LCGI Algorithm and the GCGI Algorithm to propose a Leader-Group Balance (LGB) algorithm. As shown in Algorithm 3, in each round LGB uses LCGI and GCGI to calculate the highest score $LS$ and $GS$ of the vertex and group respectively. If $LS$ is higher, it uses Algorithm 1 to get the combinatorial edge insertion. On the contrary, it uses Algorithm 2 to get the combinatorial edge insertion. The time complexity of the LGB depends on Algorithm 1 and Algorithm 2.

**Algorithm 3:** LGB ($G$, $b$)

```
Input  : A graph G = (V, E), a budget b
Output : D: the set of inserted edges
cost = 0, D ← ∅;
while cost ≤ b do
    compute LS according to Lines 3-15 in Algorithm 1;
    compute GS according to Lines 3-22 in Algorithm 2;
    if LS > GS then
        update D, cost, G according to Algorithm 1;
    else update D, cost, G according to Algorithm 2 ;
return D;
```

Theorem 4.8. *The time complexity of LGB is $O(b \cdot k_{max} \cdot n \cdot m)$.*

Proof. Since LGB algorithm calculates $LS$ and $GS$ at most $b$ times in the worst case, the worst time complexity should be the time complexity of calculating $LS$ and $GS$, which are $O(n \cdot m)$ and $O(n \cdot m \cdot k_{max})$ respectively. Therefore, the final overall complexity will not exceed $O(b \cdot n \cdot m \cdot k_{max})$. □

Since LGB algorithm considers the optimal solution of two greedy algorithms, it can bring higher coreness gain. Therefore LGB can be used when higher effectiveness is required. Due to the change in the greedy strategy, the running time is not the sum of the two algorithms. In fact, LGB, which combines the two greedy algorithms, will reduce the number of iterations, so the actual running time is sometimes even better than LCGI and GCGI. We will conduct more coreness gain and running time analysis in Section 5.

# 5 Experiments

**Datasets.** Experiments are conducted on 13 real-world large graphs, which can be found in SNAP [19] or Network Repository [27]. The statistics are shown in Table 1, sorted by increasing number of vertices, where abbreviations are used for each dataset, $\overline{deg}$ and $deg_{max}$

represent the average $deg(.)$ and the max $deg(.)$ respectively, and $k_{max}$ represents the max coreness among all $v \in V(G)$.

**Table 1: Datasets**

| Dataset | Abbr. | $n = \|V\|$ | $m = \|E\|$ | $\overline{deg}$ | $deg_{max}$ | $k_{max}$ |
|---|---|---|---|---|---|---|
| twitter_copen | TC | 8,580 | 473,614 | 110 | 1,516 | 582 |
| pkustk02 | PK | 10,800 | 410,400 | 76 | 155 | 71 |
| Email-Enron | EE | 36,692 | 183,831 | 10.02 | 1,383 | 43 |
| Facebook | FB | 63,731 | 817,035 | 25.64 | 1,098 | 52 |
| Gowalla | GW | 196,591 | 950,327 | 9.67 | 14,730 | 51 |
| DBLP | DB | 317,080 | 1,049,866 | 6.62 | 343 | 112 |
| Amazon | AZ | 334,863 | 925,872 | 5.53 | 549 | 6 |
| youtube | YT | 495,957 | 1,936,748 | 7.81 | 25,409 | 49 |
| Google | GG | 875,713 | 4,322,051 | 9.87 | 6,332 | 44 |
| lastfm | LF | 1,191,805 | 4,519,330 | 7.58 | 5,150 | 70 |
| pokec | PC | 1,632,803 | 30,622,564 | 37.51 | 14,854 | 47 |
| flixster | FS | 2,523,386 | 7,918,801 | 6.28 | 1,474 | 68 |
| LiveJournal | LJ | 4,847,571 | 68,993,773 | 28.46 | 20,333 | 372 |

**Settings.** All algorithms are implemented in C++ and compiled by g++ compiler at -O3 optimization level, and they are conducted on a machine with Inter(R) Xeon(R) CPU E5-2667 v4@3.20GHz processor and 256GB memory, with Ubuntu installed.

**Algorithms.** Since there is no algorithm that can directly solve the BLCM problem, to demonstrate the efficiency of our algorithm, we compare it with: (1) four heuristic algorithms; (2) EKC [36] algorithm revised to solve BLCM; and (3) an exact algorithm to explore edge insertion combinations. Additionally, we contrast it with the SOTA algorithms specialized in $k$-core maximization (KM) such as VEK [37] and FASTCM+ [30]. Detailed descriptions of all algorithms are shown in Table 2. Due to lack of space, some experiments are supplemented in Appendix D.

**Table 2: Description of Algorithms**

| Algorithm | Description |
|---|---|
| **Exact** | Enumerate all $b$ edge combinations from $\binom{\|V\|}{2} \setminus \|E\|$ and select the optimal strategy. |
| **Rand** | Randomly insert $b$ edges from $\binom{\|V\|}{2} \setminus \|E\|$. |
| **deg** | Select the vertex with the highest $deg(u)$ and lead $u$ to $k$-core, where $k$ is determined in the same way as Section 4.1.2 and the same below. |
| **deg-c** | Select the vertex with the highest $deg_c(u) = deg(u) - core(u)$ and lead $u$ to $k$-core. |
| **deg-s** | Select the vertex with the highest $deg_s(u) = \|\{v \| v \in nbr(u) \wedge (core(v) > core(u) \text{ or } core(u) = core(v) \wedge layer(u) > layer(v))\}\|$ and lead $u$ to $k$-core. |
| **EKC** | A greedy edge-enumeration algorithm for $k$-core maximization algorithm [36]. We extend it to solve the BLCM problem. |
| **VEK** | A greedy vertex-enumeration algorithm for $k$-core maximization algorithm [37]. |
| **FASTCM+** | The state-of-the-art algorithm for $k$-core maximization algorithm [30]. |
| **LCGI** | A greedy edge insertion strategy based on leader-followers structure in Algorithm 1. |
| **GCGI** | A greedy edges insertion strategy based on group-centric structure in Algorithm 2. |
| **GCGI-R** | GCGI without GroupReduction. |
| **GCGI-E** | GCGI without GroupExpansion. |
| **GCGI-R-E** | GCGI without GroupReduction and Group Expansion. |
| **LGB** | A combined greedy edge insertion strategy to balance LCGI and GCGI in Algorithm 3. |

## 5.1 Effectiveness

We compare the effectiveness of the proposed algorithms with four heuristic algorithms, the modified EKC algorithm, and the exact algorithm in small graphs respectively.

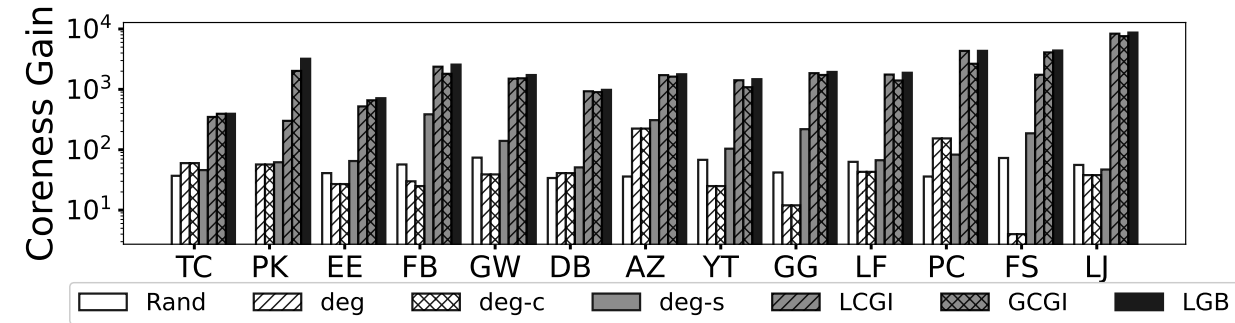

**Figure 3: Coreness Gain from Different Heuristics, $b$=50.**

**1. Comparison with heuristic algorithms.** We compare with the heuristic algorithms on 13 datasets and the results are depicted in Figure 3. It can be seen that the proposed LCGI, GCGI, and LGB algorithms all achieve remarkably higher coreness gain improvement than the four heuristic algorithms. LGB performs slightly better than LCGI and GCGI for it switches between the two policies adatively. Simple degree-based edge selections perform even worse than the random edge insertion in some datasets, showing the BLCM problem's non-intuitiveness.

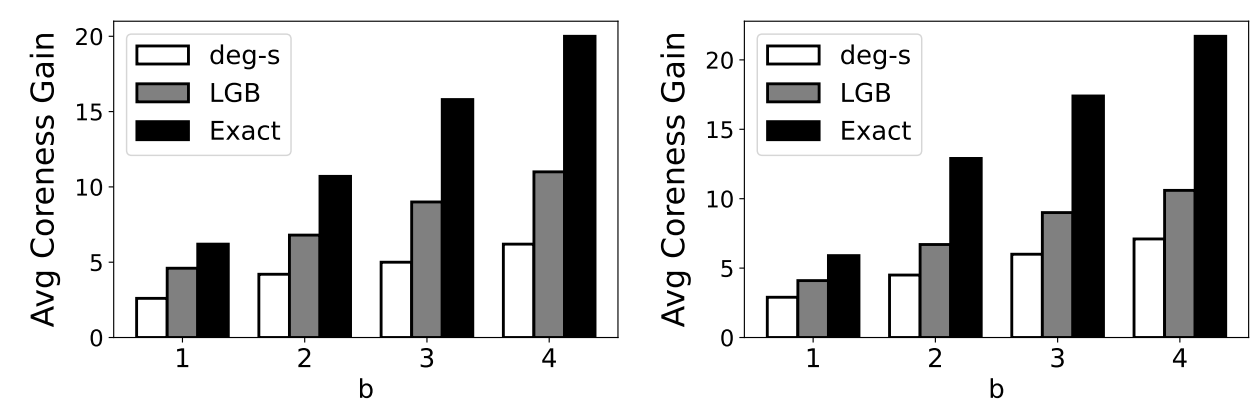

**(a) Gowalla** **(b) Facebook**

**Figure 4: LGB vs. Exact and deg-s on Coreness Gain.**

**2. Comparison with Exact algorithm.** Due to the complexity of combinatorial search, Exact algorithm is only tested on the Gowalla and Facebook datasets after vertex sampling. Specifically, we randomly select a vertex and iteratively add its neighbors until the graph contains 50 vertices. For each dataset, we take 10 small graphs and calculate the average of the final results. We additionally select the deg-s algorithm, which performs better among the heuristic algorithms, for comparison. The experimental outcomes are shown in Figure 4. Comparing with the coreness gain of the Exact algorithm, LGB still provides a substantial coreness gain, and the efficiency comparison will be shown in Section 5.2. In addition, it can be seen that the coreness gain of the deg-s algorithm is far behind that of the Exact algorithm and is still much less than LGB, which further illustrates the effectiveness of our algorithm.

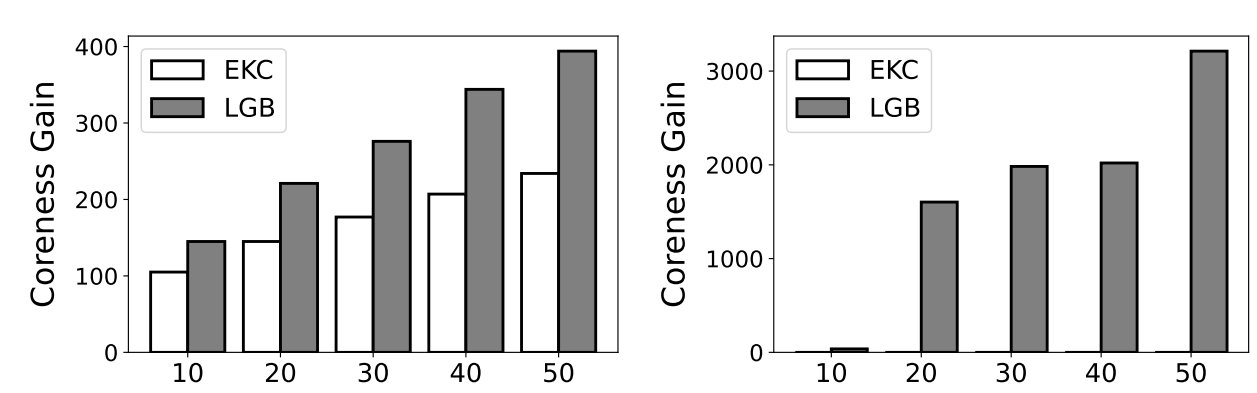

**(a) twitter_copen** **(b) pkustk02**

**Figure 5: LGB vs. EKC on Coreness Gain.**

**3. Comparison with the extended EKC algorithm.** To adapt the EKC algorithm for the BLCM problem, we extend its edge traversal range to $\binom{|V|}{2} \setminus |E|$, so it can carry out the corness maximization

task. However, from the comparison results shown in Figure 5, we can see that due to EKC's inability to explore the benefits of edge combinations, the coreness gain achieved by the EKC algorithm on twitter_copen and pkustk02 is less than LGB, even failing to yield any coreness gain on pkustk02.

## 5.2 Efficiency

In efficiency comparison, we conduct experiments using four algorithms: LCGI, GCGI, LGB and extended EKC. We set $b$ to 50.

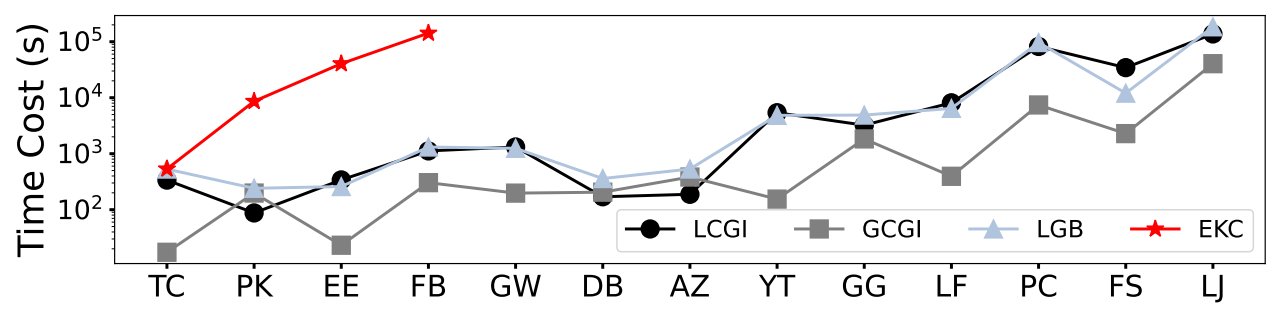

**Figure 6: Time Cost from Different Algorithms, $b$=50.**

**Overall Performance.** The experimental results are shown in Figure 6. The EKC algorithm cannot be fully displayed because it takes too long to run ($> 10^5 s$) on bigger datasets after FB or runs out of memory. As the graph size increases, the time-costs of the three proposed algorithms show a slow increasing trend. GCGI usually takes less time. Though LGB usually takes a longer time, it does not merely sum up the time costs of LCGI and GCGI. This is primarily because the fusion of the two greedy strategies results in alterations of the running rounds. In addition, the time of the extended EKC algorithm increases quickly as the data scale increases. Considering the corness gains achieved by our proposed algorithms, they achieve impressive corness gain improvements while using affordable time costs.

**Table 3: Comparison of Time Cost (s) between LGB, Exact, and EKC on Different Datasets**

| b | Gowalla | | Facebook | | b | pkustk02 | | Email-Enron | |
|---|---|---|---|---|---|---|---|---|---|
| | Exact | LGB | Exact | LGB | | EKC | LGB | EKC | LGB |
| 1 | 0.001 | **0.001** | 0.002 | **0.001** | 10 | 1926.4 | **116.2** | 9487.4 | **56.2** |
| 2 | 0.360 | **0.001** | 0.711 | **0.001** | 20 | 3513.1 | **156.7** | 17962.1 | **127.7** |
| 3 | 104.74 | **0.001** | 208.01 | **0.001** | 30 | 5397.4 | **156.9** | 26691.5 | **159.1** |
| 4 | 28053.6 | **0.001** | 57627.7 | **0.001** | 40 | 7145.6 | **239.6** | 31843.5 | **230.6** |
| 5 | >100000 | **0.001** | >100000 | **0.001** | 50 | 8619.0 | **242.3** | 40521.3 | **258.9** |

**Comparison with the Exact and the extended EKC algorithms.** The experimental settings for comparing with Exact and EKC Algorithms are the same as the previous comparison of coreness gain, and the final time cost is shown in Table 3. LGB has a remarkable improvement in time compared to Exact. The Exact algorithm requires huge time costs even on small-scale graphs and small $b$. Therefore, although LGB loses a certain degree of coreness gain, it is highly efficient in larger-scale graphs, so the reduction in coreness gain is acceptable.

In comparison with extended EKC, the LGB algorithm has a significant improvement in time. As the graph size increases further, such as from pkustk02 to the Email-Enron dataset, the time complexity of EKC increases exponentially, which makes it unable to handle large-scale datasets.

## 5.3 Comparison with other KM algorithms

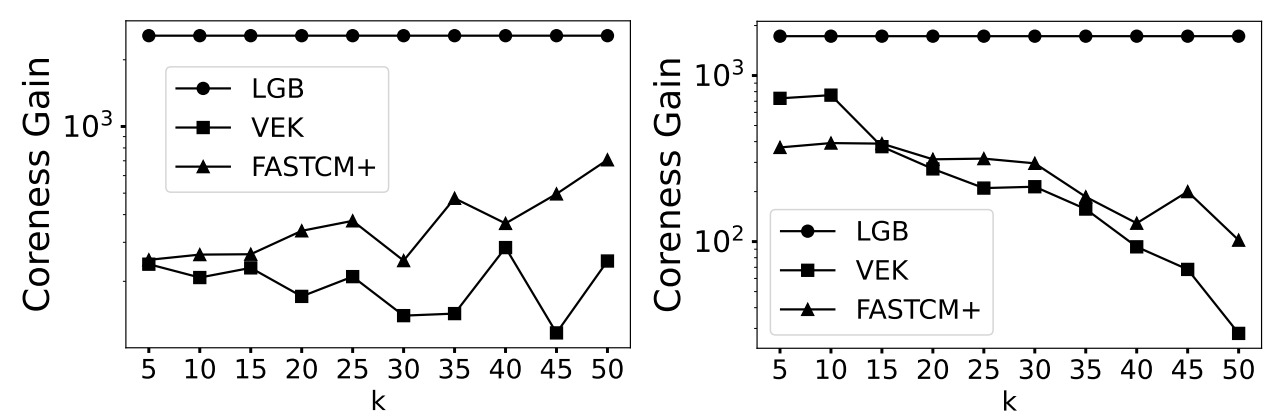

**(a) Facebook** **(b) Gowalla**

**Figure 7: Coreness Gain on different $k$.**

VEK and FASTCM+ are efficient algorithms used to address the KM problem. We compare the sum corness gains achieved by these KM algorithms and the LGB algorithm. We focus on the Facebook and Gowalla datasets with $b = 50$, and $k$ varying from 5 to 50. The results are illustrated in Figure 7. The experiments reveal that the effectiveness of VEK and FASTCM+ varies significantly when different values of $k$ are set. However, the coreness gain they achieve is both less than the LGB algorithm. Additionally, compared to the KM problem, which needs to manually specify $k$, the LGB algorithm eliminates this requirement, reducing the manual effort involved.

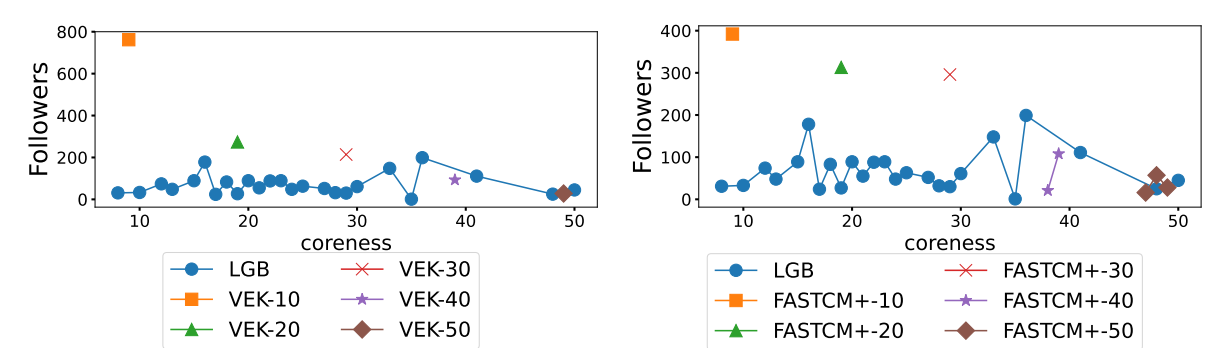

**(a) VEK** **(b) FASTCM+**

**Figure 8: Distribution of followers on Coreness in Gowalla.**

In addition, we conduct further experiments to explore the coreness distribution of followers in the KM and BLCM problems. We applied the VEK and FASTCM+ algorithms with different $k$ values (VEK-$k$ and FASTCM+$-k$ denote the set of the $k$) on Gowalla datasets and compared them with LGB. The results are shown in Figure 8. The followers of LGB are distributed among vertices with different coreness, whereas VEK-$k$ and FASTCM+$-k$ are concentrated in the vertex with coreness $k - 1$. The diversity of followers in the BLCM problem is more conducive to real-world scenarios and can better enhance average user engagement.

## 6 Conclusions

From the perspective of enhancing social network average user engagement through combinatorial edge insertions, this paper introduces the BLCM problem, proves the problem is NP-hard and APX-hard. We further prove the coreness gain function is not submodular. We propose efficient methods to evaluate the cascaded coreness improvements of two local combinatorial strategies and provide solutions to the key problem of evaluating the cascading effects. Based on this, we then propose three efficient combinatorial edge insertion strategies: LCGI, GCGI and LGB. We prove the polynomial time complexity of LCGI, GCGI and LGB. Experiments conducted on 13 real-world datasets highlight their practical utility, efficiency and effectiveness over existing approaches.

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

## A PROOFS OF THEOREMS

**Proof of Theorem 2.1:** We prove that the BLCM problem is NP-hard by employing a polynomial reduction from the Maximum Coverage (MC) problem [16] to our BLCM problem. Consider an instance of the decision problem of the MC problem. Given $B$, a budget $b$, a set of $d$ elements $\{e_1, ..., e_d\}$, and $c$ subsets of $\{e_1, ..., e_d\}$, $T_1, ..., T_c$, the decision problem of MC problem asks whether there exists $b$ subsets such that $|\bigcup_{1\le j\le b} T_{i_j}| \ge B$. Now we construct a

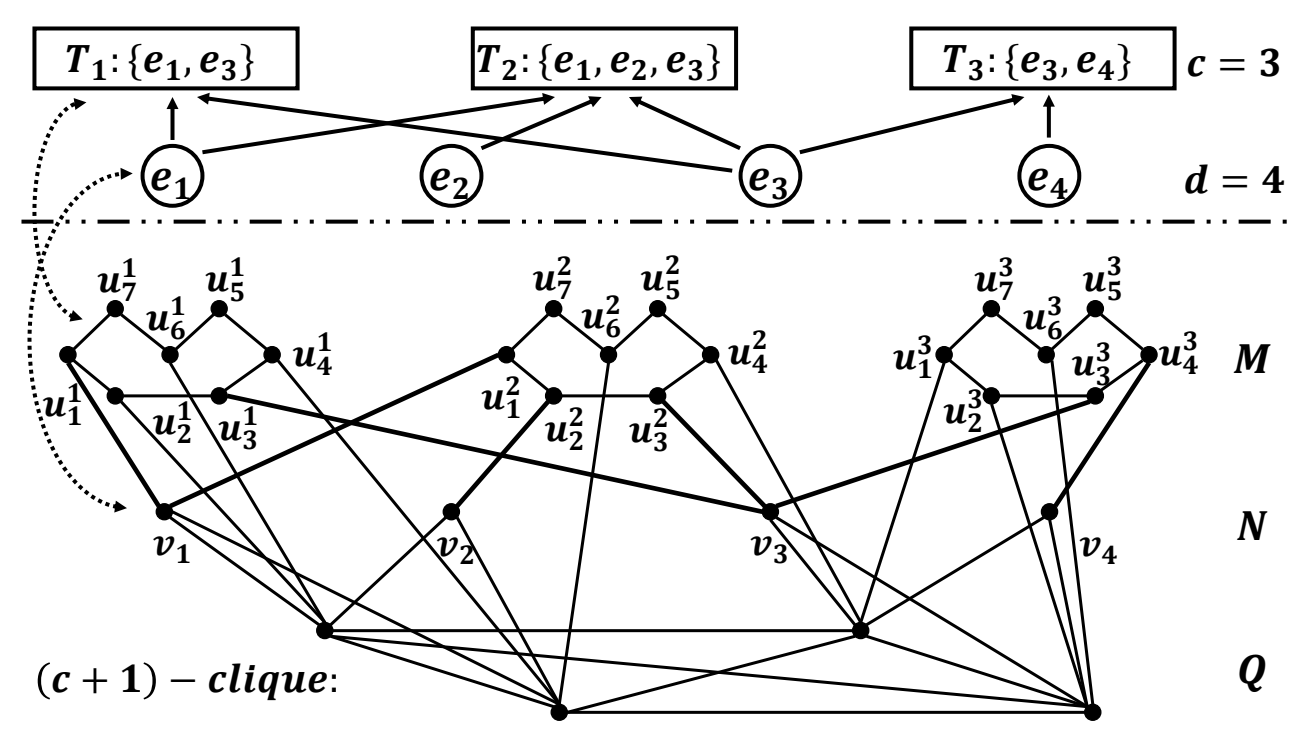

**Figure 9: Construction Example for Hardness Proofs.**

corresponding instance of the BLCM problem on a graph $G$. The graph $G$ contains three parts: $M$, $N$, and $Q$. The part $M$ contains $c$ set of vertices where each set has $d+3$ vertices, i.e., $M = \bigcup_{1\le i\le c} M_i$ where $M_i = \bigcup_{1\le j\le d+3} u_j^i$. For every $M_i$ in $M$, we connect $u_j^i$ and $u_{j+1}^i$ by an edge for every $j \in [1, d+2]$, and we also connect $u_1^i$ and $u_{d+3}^i$. The part $N$ contains $d$ vertices, i.e., $N = \bigcup_{1\le i\le d} v_i$. The part $Q$ is a $(c+1)$-clique where every two vertices of the $(c+1)$ vertices are adjacent. For every $i$ and $j$, if $e_j \in T_i$ in the MC instance, we add an edge between $v_j$ and $u_j^i$. In Figure 9, these edges are marked in bold. For every $M_i$, we add edges between every vertex in $M_i$ and the

vertices in $Q$ so that every vertex in $M_i \backslash \{u^i_{d+1}, u^i_{d+3}\}$ has degree $c$ and every vertex in $\{u^i_{d+1}, u^i_{d+3}\}$ has degree $c-1$. We add $c-1$ edges between $v_i$ and the vertices in $Q$ for every $i \in [1, d]$. The coreness of each $u^i_j \in M_i$ is $c-1$. Similarly, the coreness of each $v_j \in N$ is also $c-1$. The coreness of every vertex in $Q$ is $c$. Figure 9 shows a construction example from 3 sets and 4 elements. The decision problem of the BLCM problem is whether there exists an edge set $D \subseteq E(K_n) \backslash E(G)$ with $b$ edges such that $g(D, G) \geq b(d+3) + B$.

Now we prove that the decision problem of the MC problem has a “yes" answer if and only if the decision problem of the Corness Maximization problem has a “yes" answer.

Suppose there is a “yes" answer for the decision problem of the MC problem, that is, there is a $b$ subsets, $T_{i_1}, ..., T_{i_b}$ such that $\{e_{j_1}, ..., e_{j_B}\} \subseteq T_{i_1} \cup T_{i_2} \cup ... \cup T_{i_b}$. We set $D = \{(u^{i_1}_{d+1}, u^{i_1}_{d+3})$ ,..., $(u^{i_b}_{d+1}, u^{i_b}_{d+3})\}$. After adding $D$ to $G$, the coreness of each vertex in $M_{i_1}, ..., M_{i_b}$ increases by one; the coreness of $v_{j_1}, ..., v_{j_B}$ increases by one. Therefore, the coreness gain $g(D, G) \geq b(d+3) + B$, i.e., the decision problem of the BLCM problem has a “yes" answer.

Suppose there is a “yes" answer for the decision problem of the BLCM problem, that is, there is an edge set $D \subseteq E(K_n) \backslash E(G)$, $|D| = b$, such that $g(D, G) \geq b(d+3) + B$. In graph $G$, the coreness gain of adding any edge in $E(K_n) \backslash E(G)$ is not greater than $d + 3 + min_{1 \leq i \leq b} |T_i|$. To meet the condition that the total coreness gain of adding the $|D|$ edges is not less than $b(d+3) + B$, the coreness gain of adding any edge in $D$ is not less than $d + 3 + max_{1 \leq i \leq b} |T_i|$. Therefore, all edges in $D$ must be between $u^i_{d+1}$ and $u^i_{d+3}$, i.e., $D = \{(u^{i_1}_{d+1}, u^{i_1}_{d+3})$ ,..., $(u^{i_b}_{d+1}, u^{i_b}_{d+3})\}$. We select $T_{i_1}, ..., T_{i_b}$, and it is easy to know $|\bigcup_{1 \leq j \leq b} T_{i_j}| \geq B$. Therefore, there is a $b$ subsets, $T_{i_1}, ..., T_{i_b}$ such that $\{e_{j_1}, ..., e_{j_B}\} \subseteq T_{i_1} \cup T_{i_2} \cup ... \cup T_{i_b}$, i.e., the decision problem of the MC problem has a “yes" answer. Thus, the theorem is proved.

**Proof of Theorem 2.2:** We reduce from the MC problem using a reduction similar to that in the proof of Theorem 2.1. For any $\epsilon > 0$, the MC problem cannot be approximated in polynomial time within a ratio of $(1 - 1/e + \epsilon)$, unless P = NP [14]. Let $k$ be an arbitrarily large constant. In the construction of $G$, Q is a $k$-clique; and every $v_i$ is attached by a loop of $k$ vertices where each vertex is connected to Q by $c-2$ edges except $v_i$. Let $\gamma > 1 - 1/e$, if there is a solution with $\gamma$-approximation on the coreness gain for the BLCM problem, there will be a $\lambda$-approximate solution on optimal element number for the MC problem, where $\lambda = \gamma + \frac{(\gamma-1) \times b(d+3)}{k \times f}$ and $f$ is the coreness gain of the BLCM problem. Thus, the theorem is proved. The BLCM problem is APX-hard.

Lemma A.1. *By leading $core(x) \to k$, any vertex $u \in V(G) \setminus x$ can increase its coreness by at most 1,*

**Proof of Lemma A.1:** We prove it by contradiction. Suppose there is a vertex $u_0 \in V(G) \setminus x$ with coreness increasing from $k'$ to $k^*$ after leading $x$ to $k$-core and $k^* - k' > 1$. Then after leading $x$ to $k$-core, $u_0$ must be contained in $Q = C^{k^*}(G + E_{x,k})$ where $G + E_{x,k} = (V(G), E(G) \cup E_{x,k})$ and $E_{x,k}$ is a set of the corresponding edges for leading $x$ to $k$-core. For $\forall v \in Q$, it must have $deg(v, Q) \geq k^*$ and if we delete $x$ and $E_{x,k}$, we can get $deg(v, Q - E_{x,k} - \{x\}) \geq k^* - 1$, where $Q - E_{x,k} - \{x\} = (V(Q) \setminus \{x\}, E(Q) \setminus E_{x,k})$. Thus, $(Q - E_{x,k} - \{x\}) \subseteq C^{k^*-1}$. Since $u_0 \in Q$, we have $u_0 \in C^{k^*-1}(G)$, i.e., $k' = k^* - 1$, thus $k^* - k' = 1$, which contradicts with $k^* - k' > 1$. The theorem is proved.

**Proof of Theorem 3.1:** We denote $G^*$ as the graph after leading $core(x) \to k$, and $G'$ as the graph after anchoring $x$, i.e., setting $deg(x) = +\infty$ but not inserting edges. 1) Since the impact of anchoring $x$ must cover the effect of leading $core(x) \to k$, we can get $F(x, k) \subseteq I(x, +\infty)$. 2) We then prove that $I_k(x, +\infty) \cap F(x, k) = \emptyset$. Since $\forall v \in I_k(x, +\infty)$, we have $core(v, G) \geq k$. While $\forall v \in F(x, k)$, $core(v, G) \leq k - 1$. So, we have $I_k(x, +\infty) \cap F(x, k) = \emptyset$. 3) We then prove $I(x, +\infty) \backslash I_k(x, +\infty) \subseteq F(x, k)$. $\forall v \in I(x, +\infty) \backslash I_k(x, +\infty)$, we prove that $v \in F(x, k)$, i.e. $core(v, G^*) = core(v, G) + 1$. For $\forall v \in I(x, +\infty) \backslash I_k(x, +\infty)$, $core(v, G') = core(v, G) + 1$. We denote $I^{=}_{core(v,G)}(x, +\infty)$ as the number of followers whose coreness is equal to $core(v, G)$ before anchoring $x$. Then, $|nbr(v) \cap C^{core(v,G)+1}(G)| + |nbr(v) \cap I^{=}_{core(v,G)}(x, +\infty)| \geq core(v, G') = core(v, G) + 1$. The equation holds because $v$ must have $core(v, G) + 1$ neighbors in $C^{core(v,G)+1}(G')$. We set $F^{=}_{core(v,G)}(x, k)$ as the vertices whose coreness is equal to $core(v, G)$ among $F(x, k)$. Then it is easy to get $F^{=}_{core(v,G)}(x, k) = I^{=}_{core(v,G)}(x, +\infty)$, so we have $|nbr(v) \cap C^{core(v,G)+1}(G)| + |nbr(v) \cap F^{=}_{core(v,G)}(x, k)| \geq core(v, G')$ $= core(v, G) + 1$. Therefore $v$ also has $core(v, G) + 1$ neighbors in $C^{core(v,G)+1}(G^*)$, so $core(v, G^*) \geq core(v, G) + 1$. Based on the Lemma A.1, we have $core(v, G^*) \leq core(v, G) + 1$, so $core(v, G^*) = core(v, G) + 1$. Therefore, we have $I(x, +\infty) \backslash I_k(x, +\infty) \subseteq F(x, k)$.

Combining the above three parts, the theorem is proved.

**Proof of Theorem 3.4:** The proof of this Theorem can be easily extended from Theorem 4.14 in [21], so we will not explain it in detail.

**Proof of Theorem 3.5:** Let $K = core(v, G)$ and we divide $nbr(v)$ into three sets: $P_v = \{w | w \in nbr(v) \wedge core(w) < K\}$; $Q_v = \{w | w \in nbr(v) \wedge K = core(w) \wedge layer(v) > layer(w)\}$; $N_v = \{w | w \in nbr(v) \wedge w \notin P_v \cup Q_v\}$. (1) If $\exists x \in \mathcal{G}(u), x \in P_v \cup Q_v$, then $x \rightsquigarrow v$ is already an upstair path. (2) If there is no case (1) and $\exists x \in \mathcal{G}(u)$, $x \in N_v$, we set the graph after promoting $\mathcal{G}(u)$ as $G_{\mathcal{G}(u)}$. According to the core decomposition, $core(v, G_{\mathcal{G}(u)}) = K$ because vertices in $\mathcal{G}(u)$ is always deleted earlier in core decomposition and does not affect the deletion order of $v$. So $v$ is not a follower of $\mathcal{G}(u)$, which contradicts that $v$ is a follower. (3) If $\forall x \in \mathcal{G}(u)$, $x \notin P_v \cup Q_v \cup N_v$, $v$ must have a neighbor $v_0 \in Q_v \cap C^{K+1}(G_{\mathcal{G}(u)})$; otherwise $core(v, G_{\mathcal{G}(u)}) = K$ same as the case (2). So if $v_i$ want to be a follower of $\mathcal{G}(u)$, $v_i$ must have a neighbor $v_{i+1} \in Q_{v_i} \cap C^{K+1}(G_{\mathcal{G}(u)})$ or $v_{i+1} \in \mathcal{G}(u)$. Recursively, we can get that there exists a path $(x, ..., v)$ is an upstair path, i.e., $\exists x \in \mathcal{G}(u), x \rightsquigarrow v$. The theorem is proved.

**Proof of Theorem 4.1:** The proof of this Theorem can be easily extended from Theorem 4.15 in [21], so we will not explain it in detail.

**Proof of Theorem 4.5:** Let $k' = core^*(v) + 1$ , if $deg(v, C^{k'}(G_{\mathcal{G}(u)}) < k'$, we have $core(v, G_{\mathcal{G}(u)}) \leq k' - 1$. We will prove $d^+(v)$ computes the upper bound of $deg(v, C^{k'}(G_{\mathcal{G}(u)})$. We classify all the $w = nbr(v) \wedge w \notin \mathcal{G}(u)$ into different sets. $S = \{w | Survive(w) > 0 \wedge core^*(w) \geq core^*(v) + 1\}$ are considered in $d^+_s(v)$ unless they are never pushed into $H$. $U = \{w | Survive(w) = 0 \wedge core(w) = core^*(v)\}$ are considered in $d^+_u(v)$ unless they are never pushed into $H$. $R = \{w | Survive(w) > 0 \wedge core^*(w) = core^*(v)\}$ are considered

in $d_r^+(v)$ unless they are never pushed into $H$. $P = \{w|core(w) \geq core^*(v) + 1 \wedge Survive(w) = 0\}$. Then $P \cup \{w|w \in nbr(u) \wedge w \in \mathcal{G}(u)\}$ are considered in $d_>(v)$. $N_1 = \{w|Survive(w) = 0 \wedge core(w) < core^*(v)\}$ and $N_2 = \{w|Survive(w) > 0 \wedge core^*(w) < core^*(v)\}$ can not contribute to the $deg(v, C^{k'}(G_{\mathcal{G}(u)})$. So $d^+(v)$ is the upper bound of $deg(v, C^{k'}(G_{\mathcal{G}(u)})$. Thus, if $d^+(v) < k'$, $core(v, G_{\mathcal{G}(u)}) \leq core^*(v)$. The theorem is proved.

# B Detailed Implementation of the Algorithm

In this section, we provide detailed pseudocode for the previous algorithms.

## B.1 Layer Decomposition Algorithm

We design the layer decomposition algorithm to get the layer number of vertices as shown in Algorithm 4.

**Algorithm 4:** Layer Decomposition ($G, k_{max}$)

```
Input  : G = (V, E), k_max: max coreness for v ∈ G
Output : layer(.) for ∀v ∈ G
1 for k from 1 to k_max do
2     Q ← C^k(G), i ← 1, P = {v|deg(v, Q) < k + 1 ∧ v ∈ Q};
3     while P ≠ ∅ do
4         foreach v ∈ P do layer(v) = i;
5         i + +, Q = Q \ P, P = {v|deg(v, Q) < k + 1 ∧ v ∈ Q};
6 return layer(.);
```

## B.2 FindFollowers Algorithm

**Algorithm 5:** Shrink ($u$)

```
Input  : the vertex u for degree check
1 foreach survived v ∈ nbr(u) ∧ v ≠ x ∧ core(v) = core(u) do
2     d⁺(v) = d⁺(v) − 1;
3     if d⁺(v) < core(v) + 1 then T ← v ;
4 foreach v ∈ T do
5     v is set discarded, Shrink(v);
```

The cascade update effect is simulated using Algorithm 5. The **FindFollowers** process is shown in Algorithm 6. The time complexity of Algorithm 6 is $O(m)$, as edge traversal occurs at most three times during degree checking, pushing neighbors into $H$, and invoking the Shrink function.

**Algorithm 6:** FindFollowers ($x, G$)

```
Input  : A graph G = (V, E), the leader x
Output : I(x, +∞)
1  H ← ∅, x is set survived;
2  foreach v ∈ nbr(x) do
3      if core(v) > core(x) or v ∈ cel^>(u) then
4          H.push({{core(v), layer(v)}, v});
5  while H ≠ ∅ do
6      u ← H.pop(), Compute d⁺(u);
7      if d⁺(u) ≥ core(u) + 1 then
8          u is set survived;
9          foreach v ∈ nbr(u) do
10             if v ∈ cel^>(u) ∧ v ∉ H then
11                 H.push({{core(v), layer(v)}, v});
12     else u is set discarded, Shrink(u) ;
13 return survived vertices \ {x};
```

## B.3 GroupReduction Algorithm

**GroupReduction** is shown in Algorithm 7. The $gr(.)$ and $gc(.)$ values are initially calculated for each vertex $u$ (Line 1), and then it is assessed whether $u$ will be removed from $\mathcal{G}(v)$ according to $gv(.)$ (Lines 2-5). If $u$ will be removed, the sub-function **GroupShrink** (Lines 6-9) is called. This function traverses the $nbr(u)$ that are still in $\mathcal{G}(v)$, updates the corresponding $gr(.)$ and $gc(.)$, and determines whether these vertices should be removed from the $\mathcal{G}(v)$.

**Algorithm 7:** GroupReduction ($G$, $\mathcal{G}(v)$)

```
Input  : A graph G = (V, E), a group 𝒢(v) centered on v
1 foreach u ∈ 𝒢(v) do compute gr(u) and gc(u) ;
2 foreach u ∈ 𝒢(v) do
3     if gc(u) − gr(u) < 0 then
4         Remove u from 𝒢(v);
5         GroupShrink(u, gr(.), gc(.));
6 Procedure GroupShrink(u, gr(.), gc(.))
7     foreach w ∈ nbr(u) ∧ w ∈ 𝒢(v) do
8         gr(w)+ = 1, gc(w)− = 1;
9         if gc(w) − gr(w) < 0 then Similar to Lines 4-5 ;
```

## B.4 GroupExpansion Algorithm

The Algorithm 8 illustrates the process of **GroupExpansion**. Initially, it computes $er(.)$ and $ec(.)$ for the vertices within one-hop of $\mathcal{G}(v)$ and uses array *visited(.)* to mark whether they have been

**Algorithm 8:** GroupExpansion ($G$, $\mathcal{G}(v)$)

```
Input  : A graph G = (V, E), a group 𝒢(v), gr(.)
1  visit(.) = 0, K_E = core(v) + 1;
2  foreach u ∈ 𝒢(v) do
3      foreach w ∈ nbr(u) ∧ core(w) < K_E ∧ visit(w) = 0 do
4          compute er(w) and ec(w), visit(w)=1;
5          if ec(w) − er(w) ≥ 0 then
6              Add w to 𝒢(v);
7              foreach x ∈ nbr(w) ∧ x ∈ 𝒢(v) ∧ gr(x) > 0 do
8                  gr(x)− = 1;
9                  if gr(x) = 0 then
10                     foreach q ∈ nbr(x) ∧ visit(q)=1 do
11                         ec(q)− = 1;
12             GroupAmplify(w, er(.), ec(.));
13 Procedure GroupAmplify(w, er(.), ec(.))
14     foreach u ∈ nbr(w) ∧ u ∉ 𝒢(v) ∧ visit(u) = 1 do
15         er(u)− = 1, ec(u)+ = 1;
16         if ec(u) − er(u) ≥ 0 then Similar to lines 6-12 ;
```

visited (Line 4). If $ev(.) \geq 0$, the vertex is added to $\mathcal{G}(v)$ and the algorithm calls the subfunction **GroupAmplify** to reevaluate the vertices that have been visited but do not satisfy $ev(.) \geq 0$ (Lines 5-12). If they currently satisfy $ev \geq 0$, they are added to the group (Lines 13-16). And we need to update the $gr(.)$ of the vertices in the $\mathcal{G}(v)$ (Lines 8-11), because only the vertices in $\mathcal{G}(v)$ with $gr(.) > 0$ can participate in the calculation of $ec(.)$.

**Complexity Analysis.** In Algorithm 8, since each vertex will only be calculated once $ec(.)$ and $er(.)$, GroupAmplify will be called $n$ times in the worst case, and Line 15 will only happen $O(\sum_{v \in V(G)} deg(v)) = O(m)$ times in the worst case. Line 8 can be called at most (including in GroupAmplify) $O(n \cdot gr_{max}) < O(n \cdot k_{max})$ times Where $gr_{max}$ is the possible maximum value of $gr(.)$. The case of $gr(x) = 0$ happens at most once for each vertex, so the time complexity of Line 11 is also $O(m)$. Therefore, the final time complexity is $O(m + n \cdot k_{max})$.

## B.5 GroupGain Algorithm

**Algorithm 9:** GShrink ($v$, $\mathcal{G}(u)$)

```
Input   : the vertex v for degree check, group 𝒢(u)
foreach survived w ∈ nbr(v) with w ∉ 𝒢(u) do
    if core*(v) + 1 = core*(w) then
        d⁺(w) = d⁺(w) − 1;
        if d⁺(w) < core*(w) then
            Survive(w)− = 1, T ← w;
foreach w ∈ T do
    w is set discarded, remove w from vis;
    GShrink(w, 𝒢(u));
```

**Algorithm 10:** GroupGain ($G$, $\mathcal{G}(u)$)

```
   Input  : A graph G = (V, E), a group 𝒢(u)
   Output : Ggain: the coreness gain of 𝒢(u)
1  H = ∅, Survive = ∅, vis = ∅, FG = ∅ ;
2  Ggain = 0, K ← core(u, G) + 1, epoch_k ← 0 ;
3  foreach v ∈ 𝒢(u) do
4      H.push({{core(v), layer(v)}, v});
5  while H ≠ ∅ do
6      vk ← H.top().key;
7      if vk ≠ epoch_k then
8          push vis into H, vis ← ∅, epoch_k = vk;
9          set vertices in vis as reexplored;
10     v ← H.top().value, H.pop();
11     if v ∈ 𝒢(u) then
12         v is set survived, Insert (v, K − core(v)) into FG;
13         foreach w ∈ nbr(v) ∧ w ∉ H ∧ w is unexplored do
14             if w ∈ cel^>(v) or K > core(w) > core(v) then
15                 H.push({{core(w), layer(w)}, w});
16         continue;
17     compute d⁺(v);
18     if d⁺(v) ≥ core*(v) + 1 then
19         foreach w ∈ nbr(v)∧ w is unexplored ∧w ∉ H do
20             if v is unexplored ∧w ∈ cel^>(v) then
21                 H.push({{core(w), layer(w)}, w});
22             if v is survived ∧core*(v) = core(w) then
                    H.push({{core(w), layer(w)}, w}) ;
23         v is set survived, Survive(v)++, Insert v into vis;
24     else v is set discarded, GShrink(v, 𝒢(u)) ;
25 foreach v ∈ V(G) ∧ Survive(v) > 0 do
26     Insert (v, Survive(v)) into FG;
27 Ggain = Σ_{(v,k)∈FG(𝒢(u))} k;
28 return Ggain;
```

The Algorithm 10 details the pseudocode for computing $Ggain(.)$, where *vis* records *survive* vertices in the current hierarchy $k$, and *epoch_k* denotes the coreness value of the ongoing hierarchy (Line 1). As *epoch_k* increases, vertices in *vis* will be placed in $H$ (Lines 3-4) and marked as *reexplored*. If the current head of the queue is in $\mathcal{G}(u)$, it expands the upstairs path outwards (Lines 11-16). Otherwise, calculate $d^+(v)$, and begin the degree check. If $v$ successes (Lines 18-23), the state will be modified and expanded outwards and $v$ will be added into $vis$. When the coreness hierarchy increases, $v$ will continue to be added to $H$ and expanded. If the degree check fails (Line 24), $v$ will be marked as *discarded*, invoking the *GShrink* function. Distinct from *Shrink*, *GShrink* incorporates the effect of **Survive(.)**, determining whether the coreness of $v$ cannot increase further. Finally, we find $FG(\mathcal{G}(u))$ based on the value of $Survive(.)$ and calculate the $Ggain(\mathcal{G}(u))$ (Lines 25-27). Since the number of times a vertex is put into $H$ is bounded and does not exceed $core(u)$, the time complexity is $O(m \cdot k_{max})$.

# C Related Work

The $k$-core model [3, 28] serves as a crucial model in understanding cohesive subgraphs [7, 9, 35], with practical applications such as community discovery [11–13, 15], influence propagation [17, 20, 24], protein structure analysis [2], social network analysis [5, 12], and graph visualization [1]. The $k$-core model can evaluate the user engagement of the network [4, 25, 34]. The size of $k$-core is positively correlated with user engagement, and Linghu et al. [21, 22] further explored that coreness is better than $k$-core in reflecting user engagement. Therefore, it is necessary to improve network average user engagement from coreness improvement.

Bhawalkar et al. [4] introduced the anchor $k$-core problem. Zhang et al. [33] devised an onion-layer architecture for the anchor $k$-core problem. Laishram et al. [18] addressed this problem through core decomposition on the residual graph. Linghu et al. [21, 22] further proposed the anchor coreness problem to explore methods for enhancing global stability. Teng et al. [31] improved the anchor coreness method and proposed an Advanced Greedy Approach. Additionally, Zhang et al. [32] introduced dynamic graphs to assess the importance of vertices. However, in practical scenarios, like in social networks, such kind of vertex anchoring operation is hard to realize, i.e., increasing the vertex degree to be positive infinity without changing its connections to the other vertices is hard to realize.

**Table 4: Experiments on different variants of GCGI algorithms, $b$=50.**

| Dataset | Coreness Gain | | | | Time Cost (s) | | | |
|---|---|---|---|---|---|---|---|---|
| | GCGI-R-E | GCGI-R | GCGI-E | GCGI | GCGI-R-E | GCGI-R | GCGI-E | GCGI |
| TC | 310 | 356 | 387 | **394** | 5.28 | **5.13** | 13.39 | 16.26 |
| PK | 204 | 234 | 424 | **2020** | **88.98** | 307.14 | 150.32 | 194.84 |
| EE | 383 | 465 | 509 | **659** | 15.55 | **7.59** | 29.61 | 15.62 |
| FB | 547 | 853 | 976 | **1809** | 490.58 | **210.19** | 661.52 | 302.26 |
| GW | 639 | 941 | 1072 | **1525** | 204.01 | **114.44** | 414.37 | 187.30 |
| DB | 703 | 754 | 782 | **902** | 136.88 | **110.60** | 285.32 | 194.97 |
| AZ | 1297 | 1353 | 1531 | **1625** | 301.18 | **226.52** | 479.67 | 359.15 |
| YT | 742 | 564 | 1027 | **1086** | 629.32 | 168.18 | 1093.65 | **139.56** |
| GG | 1374 | 1601 | 1604 | **1730** | 962.54 | **760.83** | 2190.41 | 1698.55 |
| LF | 784 | 868 | 1202 | **1402** | 830.13 | **353.13** | 2195.20 | 405.44 |
| PC | 1431 | 1770 | 1638 | **2654** | 549418 | 7469.81 | 787888 | **7244.75** |
| FS | 808 | 3249 | 2822 | **4093** | 31545.2 | **800.25** | 4893.12 | 2284.88 |
| LJ | 3340 | 7042 | 4440 | **7612** | 86770.7 | **15138.2** | 224100 | 38818.4 |

Zhou et al. [36] first introduced the $k$-core maximization problem, aimed at improving network stability through edge insertion. Building upon this work, Zhou et al. [37] presented a vertex-based greedy enumeration approach. Further advancements were made by Sun et al. [30], which achieves faster performance through $k$-1-shell division and conversion. Additionally, Do et al. [10] extended the $k$-core maximization problem to hypergraphs. However, the value $k$ is hard to determine, and improving only the vertices with coreness $k-1$ can hardly reflect the overall coreness gain of the graph. Therefore, we propose a new unexplored problem with practically significance: the BLCM problem, i.e., how to enhance the engagement of nodes of all coreness as much as possible using a limited edge insertion budget.

# D Supplementary experiments

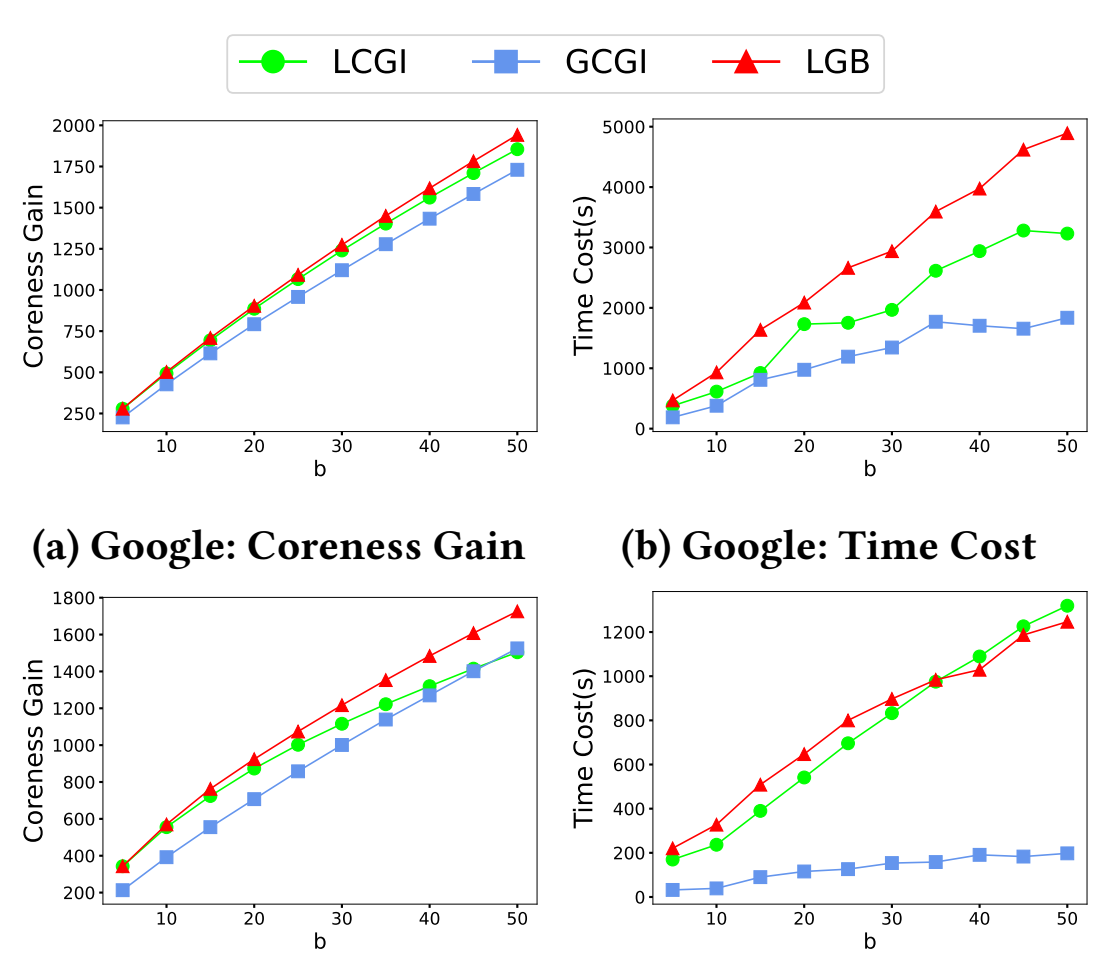

(a) Google: Coreness Gain (b) Google: Time Cost

(c) Gowalla: Coreness Gain (d) Gowalla: Time Cost

**Figure 10: Vary $b$**

## D.1 Ablation Experiments on GCGI

To assess the effectiveness of GroupExpansion and GroupReduction, we conducted experiments across 13 datasets with $b$ set to 50. The results are shown in Table 4. GCGI-R-E exhibits the poorest performance, often accompanied by longer time costs due to the absence of group pruning. GCGI-E gets better coreness gain than GCGI-R-E by removing vertices with poor performance in the group. However, in rare cases, such as the youtube dataset, vertex removal may lead to an inability to identify groups with satisfactory performance, resulting in a decrease in effectiveness. GCGI-R leads to a higher coreness gain than GCGI-R-E but is often accompanied by increased time complexity. GCGI combines the GroupReduction and GroupExpansion, which achieves higher coreness gain compared to the other three algorithms while ensuring the time cost of GCGI is not so long in most cases, making it the most suitable option for practical applications.

## D.2 Stability when Varying $b$

We conducted experiments to assess changes in coreness gain and time cost on the Gowalla and Google datasets as $b$ increases from 5 to 50, whose results are shown in Figure 10. As depicted in Figures 10a and 10c, with the increase in $b$, coreness gain exhibits a gradual upward trend across different algorithms. Moreover, it can be observed from Figures 10b and 10d that the time cost also demonstrates a slow upward trend, although this is not universally true. Since the algorithm operates based on a greedy strategy, the limitation imposed by $b$ may lead to varying greedy selection outcomes. Consequently, in different algorithms, there may be scenarios where the time decreases even as $b$ increases.

In addition, it can be seen in Figure 10d that the running time of LGB is smaller than that of LCGI, which is due to the combined greedy strategy of LGB, leading to the decrease of running epochs.

## D.3 Case Study

We conducted a case study on the soc-dolphins dataset [23], a small real-world social network consisting of 62 vertices and 159

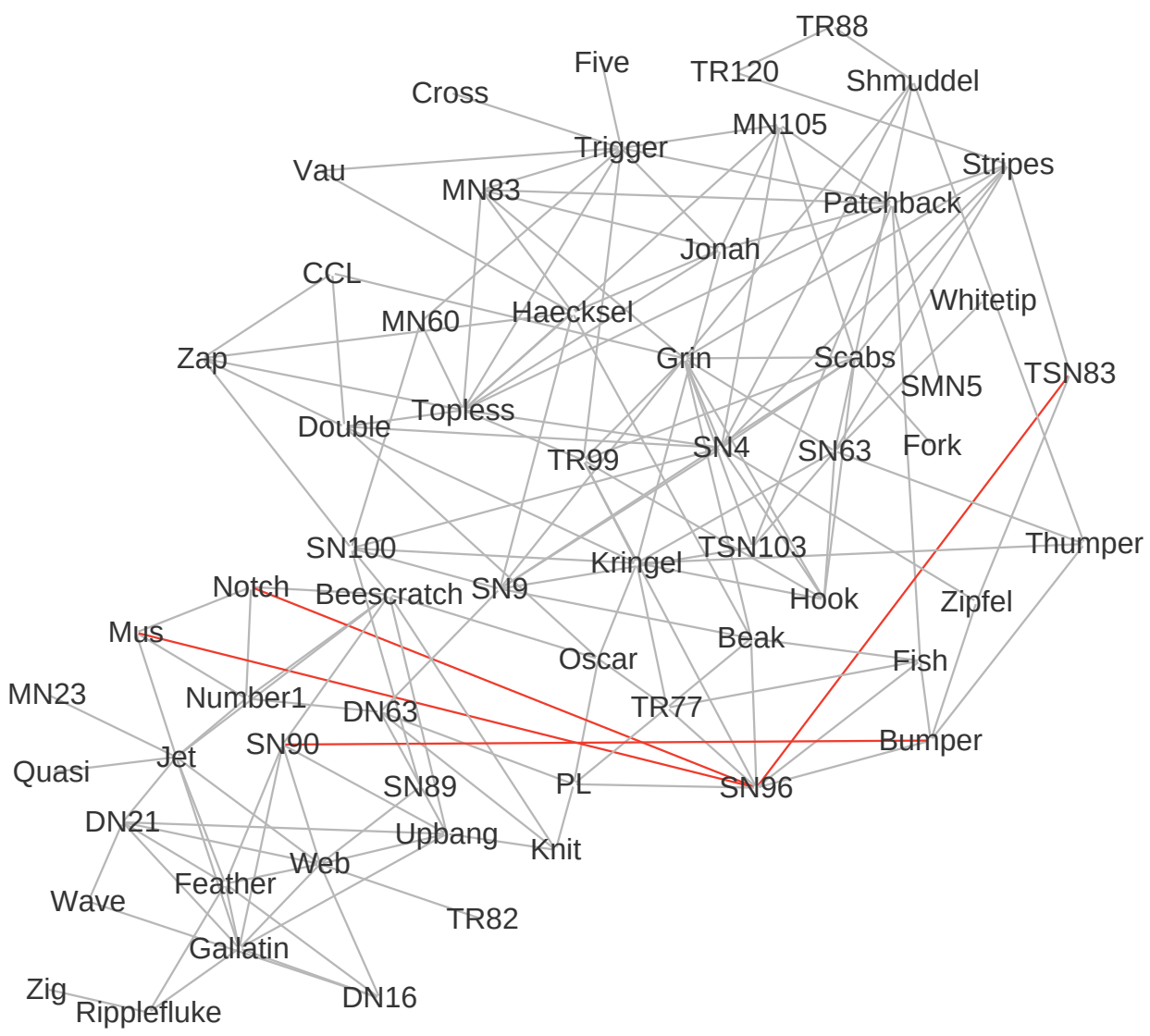

**Figure 11: A case-study on the soc-dolphins dataset.**

edges. Figure 11 illustrates the additional edges added by the LGB algorithm with a budget of $b = 4$. At this stage, the coreness gain achieved is 8. Among them, the nodes "Bumper", "Mus", "Notch", "Number1", "Shmuddel", and "Thumper" are updated from the 3-core to the 4-core, while "TSN83" and "Zipfel" are updated from the 2-core to the 3-core.