# OpenReview forum: "Coreness Maximization through Budget-Limited Edge Insertion"
_ACM.org/TheWebConf/2025/Conference — WWW 2025 Oral_

### Official Review · Reviewer_1G2e · 2024-11-05

**Novelty:** 5
**Technical Quality:** 6

**Review:**

This paper formulates a new problem, called coreness maximization under a budget, which aims to maximize the sum of all the vertex coreness values by adding b new edges into the graph. The authors prove the NP-hardness and the APX-hardness of the problem and non-submodular of the objective function. Three heuristic algorithms are proposed, LCGI, GCGI and LGB, where LGB is a hybrid version of LCGI and GCGI. Experiments on 13 real-world graphs were conducted, which verified the efficiency and effectiveness of the proposed algorithms. The paper is generally well-structured and has solid technical contributions. However, the paper still has the following drawbacks.
1. The proposed algorithms are heuristic in nature without any guarantee of the approximation ratio. Theorem 2.2 proved that the problem cannot be approximated in polynomial time within a ratio of $1-1/e+\epsilon$. However, it is unclear whether the problem can be approximated in polynomial time for a ratio that is smaller than $1-1/e$.
2. Section 3 is hard to follow, with many notations and concepts. Examples would help here.
3. According to Figure 6, the proposed heuristic algorithms are still slow in processing the three largest datasets used in the paper. Also note that these three largest datasets are not “large” in nature, as they contain less than 100 million edges.

Other minor issues.
- In computing $cost(x,k)$ by Equation (2), it also needs $F_{k-1}(x,k)$. I cannot find in the paper how $F_{k-1}(x,k)$ is computed in the algorithms.
- In Definition 3.3, for the condition at the end of the definition, are the parenthesis correctly organized, or maybe this should be reorganized to make it more clear? It seems to me that the condition is (A and B) or C.
- From Theorem 3.4 onwards, does $\leadsto$ always mean upstair paths? This is not explicitly mentioned, while Definition 3.3 uses $\leadsto$ to refer to a path.

**Questions:**

Please see comments above.

**Reviewer Confidence:**

3: The reviewer is confident but not certain that the evaluation is correct

**Scope:**

3: The work is somewhat relevant to the Web and to the track, and is of narrow interest to a sub-community

---

### Official Review · Reviewer_7FXz · 2024-11-19

**Novelty:** 5
**Technical Quality:** 6

**Review:**

This work proposes the Budget Limited Coreness Maximization problem, and shows that the problem is interesting by proving the problem is NP-hard, APX-hard, and not submodular. Efficient solutions for the problem are provided.

**Questions:**

User engagement in social networks is reflected simply by local properties, such as degree and clustering coefficient, vertex degree may exhibit a strong positive correlation with the user engagement on the Gowalla [19] dataset as well. Motivation for the proposed problem is not clear enough.

**Reviewer Confidence:**

3: The reviewer is confident but not certain that the evaluation is correct

**Scope:**

4: The work is relevant to the Web and to the track, and is of broad interest to the community

---

### Official Review · Reviewer_2pF9 · 2024-11-28

**Novelty:** 5
**Technical Quality:** 6

**Review:**

Summary:

This paper investigates the budget limited coreness maximization (BLCM) problem. Specifically, BLCM aims to improve average user engagement by inserting a limited number of edges, which is proven NP-hard / APX-hard and not submodular. To solve this problem, three edge insertion strategies (Leader-centric greedy insertion, group-centric greedy insertion, and leader-group balance insertion) are proposed and evaluated through extensive experiments.




Strength:

1.	This paper presented solid analyses of the proposed strategies for edge insertion for coreness maximization.

2.	The time complexities of the proposed three strategies are clearly demonstrated.

3.	The proposed methods have been extensively evaluated across 13 real-world large graphs against diverse baselines and the performance is validated.


Weakness:
1.	I understand that it has been proved that the BLCM problem cannot be approximated in polynomial time beyond a ratio of (1-1/e+epsilon), but I am still curious how well the proposed strategies approximate the optimal solution if possible.

2.	It incurs substantial core concepts when presenting the edge insertion strategies. It would be better understood if there are illustrations to explain the rationales.

3.	Three strategies are proposed. It would be better to include a discussion on comparing their advantages or feasibility towards various real-world applications.

**Questions:**

Please refer to the weakness part.

**Reviewer Confidence:**

2: The reviewer is willing to defend the evaluation, but it is likely that the reviewer did not understand parts of the paper

**Scope:**

3: The work is somewhat relevant to the Web and to the track, and is of narrow interest to a sub-community

---

### Official Review · Reviewer_eyiC · 2024-12-01

**Novelty:** 4
**Technical Quality:** 4

**Review:**

This paper studies the Budget Limited Coreness Maximization problem, which seeks to enhance average user engagement by inserting up to b  edges to maximize coreness gain across a graph. The authors prove that BLCM is NP-hard and APX-hard. To overcome the challenges, the paper proposes a greedy framework that combines local combinatorial optimization with global greedy search, enabling efficient and effective edge insertion. Three strategies are introduced: Leader-Centric Greedy Insertion (LCGI), Group-Centric Greedy Insertion (GCGI), and Leader-Group Balance (LGB). Experiments on real-world datasets are conducted to test the practical effectiveness and efficiency, including a case study.

**Questions:**

1.While anchoring the vertices are extensively studied, it is necessary for the paper to discuss more application scenarios of inserting edges for increased coreness gain.

2. This paper follows a similar greedy framework like existing works [21, 22] on anchored coreness maximisation problem. Can the authors highlight the unique challenges with designing algorithms for inserting edges for coreness gain?

3.  Previous works [21, 22] have investigated the anchored coreness problem and proposes efficient and effective greedy algorithms. These algorithms can be modified to handle the BLCM problem in this paper. A comparison of the effectiveness of the proposed algorithms to the methods in [21, 22] in the experiments in various settings is necessary.

4. The paper proposes a series of optimisations to improve effectiveness and efficiency. E.g., (1) Leader-Centric Greedy Insertion, (2) Group-Centric Greedy Insertion, and (3) a Leader-Group Balance. Can the authors evaluate the effect of these optimizations on varying budgets?

**Reviewer Confidence:**

4: The reviewer is certain that the evaluation is correct and very familiar with the relevant literature

**Scope:**

4: The work is relevant to the Web and to the track, and is of broad interest to the community

---

### Official Review · Reviewer_6WWA · 2024-12-03

**Novelty:** 4
**Technical Quality:** 4

**Review:**

This paper investigates the Budget-Limited Coreness Maximization (BLCM) problem, which aims to maximize the overall coreness gain of vertices in a graph by inserting up to b edges, thereby enhancing user engagement in social networks. The paper introduces three efficient combinatorial edge insertion strategies: Leader-Centric Greedy Insertion (LCGI), Group-Centric Greedy Insertion (GCGI), and a balanced approach that adaptively combines both (LGB). Experimental results on 13 real-world datasets demonstrate the superiority of the proposed methods in both coreness enhancement and computational efficiency.

Strong points:

S1. The paper provides a precise definition of the Budget-Limited Coreness Maximization (BLCM) problem and rigorously proves its computational complexity, including NP-hardness, APX-hardness, and non-submodularity.

S2. Three novel combinatorial edge insertion strategies are proposed.

S3. Extensive experiments conducted on 13 real-world datasets highlight the effectiveness and efficiency of the proposed methods in improving vertex coreness and computational performance.

Weak points:

W1: The motivation of the paper should be further enhanced.

W2: Unclear relation between BLCM and KM problems.

W3: The presentation can be improved by including concreate examples.

**Questions:**

Q1. The problem proposed in this paper is novel and requires further elaboration to emphasize its importance. For instance, specific real-world examples should be provided to demonstrate the practical relevance of the concept introduced. Without such examples, it is difficult to substantiate the significance of the proposed problem.


Q2. There exists a substantial overlap between the BLCM and KM problems. The paper only discusses the differences in their objectives but does not explain the similarities and commonalities between them. For instance, it would be useful to explore whether the two problems can be transformed into one another, and if not, to clarify the key difficulties involved in distinguishing them.

Q3. An illustrative example should be provided in Section 4.1 to clarify the concepts and enhance the understanding of the discussed methods.

Q4. The time complexity of the Layer Decomposition process is not considered in Theorem 4.2.

Q5. The first sentence of section 5.1.2, "Due to the complexity of combinatorial search, the Exact algorithm is only tested on the Gowalla and Facebook datasets after vertex sampling," presents a logical inconsistency. The choice of these two datasets is not adequately justified, and the cause-and-effect relationship is not well-established.

Q6. The paper employs a significant amount of notation, and providing a symbol table would greatly improve clarity and accessibility for readers, making it easier to follow the mathematical formulations.

Q7. Algorithm Comparison: It is unclear why only one algorithm was extended in the experiments. Comparing algorithms designed for different optimization directions is unfair. For instance, comparing the performance of LGB with VEK and FASTCM+ on KM problem is also unacceptable.

**Reviewer Confidence:**

3: The reviewer is confident but not certain that the evaluation is correct

**Scope:**

2: The connection to the Web is incidental, e.g., use of Web data or API